# Context-specific emergence and growth of the SARS-CoV-2 Delta variant

John T. McCrone[1,29], Verity Hill[1,29], Sumali Bajaj[2,29], Rosario Evans Pena[2,29], Ben C. Lambert[3], Rhys Inward[2,4], Samir Bhatt[4,5], Erik Volz[4], Christopher Ruis[6], Simon Dellicour[7,8], Guy Baele[8], Alexander E. Zarebski[2], Adam Sadilek[9], Neo Wu[9], Aaron Schneider[9], Xiang Ji[10], Jayna Raghwani[2], Ben Jackson[1], Rachel Colquhoun[1], Áine O'Toole[1], Thomas P. Peacock[11,12], Kate Twohig[12], Simon Thelwall[12], Gavin Dabrera[12], Richard Myers[12], The COVID-19 Genomics UK (COG-UK) Consortium*, Nuno R. Faria[2,4,13], Carmen Huber[14], Isaac I. Bogoch[15,16], Kamran Khan[14,16,17], Louis du Plessis[2,18,19], Jeffrey C. Barrett[20], David M. Aanensen[21], Wendy S. Barclay[11], Meera Chand[12], Thomas Connor[22,23,24], Nicholas J. Loman[25], Marc A. Suchard[26], Oliver G. Pybus[2,27,28,30✉], Andrew Rambaut[1,30✉] & Moritz U. G. Kraemer[2,28,30✉]

The SARS-CoV-2 Delta (Pango lineage B.1.617.2) variant of concern spread globally, causing resurgences of COVID-19 worldwide[1,2]. The emergence of the Delta variant in the UK occurred on the background of a heterogeneous landscape of immunity and relaxation of non-pharmaceutical interventions. Here we analyse 52,992 SARS-CoV-2 genomes from England together with 93,649 genomes from the rest of the world to reconstruct the emergence of Delta and quantify its introduction to and regional dissemination across England in the context of changing travel and social restrictions. Using analysis of human movement, contact tracing and virus genomic data, we find that the geographic focus of the expansion of Delta shifted from India to a more global pattern in early May 2021. In England, Delta lineages were introduced more than 1,000 times and spread nationally as non-pharmaceutical interventions were relaxed. We find that hotel quarantine for travellers reduced onward transmission from importations; however, the transmission chains that later dominated the Delta wave in England were seeded before travel restrictions were introduced. Increasing inter-regional travel within England drove the nationwide dissemination of Delta, with some cities receiving more than 2,000 observable lineage introductions from elsewhere. Subsequently, increased levels of local population mixing—and not the number of importations—were associated with the faster relative spread of Delta. The invasion dynamics of Delta depended on spatial heterogeneity in contact patterns, and our findings will inform optimal spatial interventions to reduce the transmission of current and future variants of concern, such as Omicron (Pango lineage B.1.1.529).

The SARS-CoV-2 pandemic has been characterized by the appearance and spread of genetically distinct virus variants that are associated with faster spread than pre-existing lineages. In May 2021, the World Health Organization (WHO) announced a new variant of concern (VOC), designated Delta. Delta became the variant primarily responsible for a wave of transmission and mortality in India in early-to-mid 2021, replacing Alpha (Pango lineage B.1.1.7) and Kappa (Pango lineage B.1.617.1) in the process[3,4]. Studies indicate that Delta has increased transmissibility[5], rates of hospitalization[6] and immune evasion[7] compared with Alpha[8,9], the variant that was previously dominant in many countries. These phenotypes are attributed to a constellation of 30 mutations across the virus genome (Supplementary Table 2) compared to the Wuhan-Hu-1 reference sequence, including: the spike mutation P681R in the furin cleavage site, which is thought to increase the speed and efficiency

with which the virus fuses with host cells[10,11]; mutation L452R in the receptor-binding domain, which is thought to reduce neutralization by antibodies[12]; and the nucleocapsid mutation R203M, which is thought to increase virion infectivity[13]. Delta disseminated rapidly from India to locations worldwide and has been detected in 174 countries as of 12 April 2022 (https://cov-lineages.org/global_report.html). Delta became the dominant lineage in the UK by mid-May 2021[14], and similar increases in frequency were observed worldwide (for example, ref. [15]).

The emergence of Delta in the UK occurred in the context of a heterogeneous landscape of prior immunity (from infection and vaccination) and non-pharmaceutical interventions (NPIs). Here we examine virus genomes generated from a random sample of all COVID-19-positive tests with PCR with reverse transcription (RT–PCR) $C_t$ values greater than 30, collected during community-based testing in England between

12 March 2021 and 15 June 2021. Our data include 52,992 Delta genomes from England with known dates and locations of sampling, representing more than 40% of all positive lateral flow and PCR tests in England during the study period (see Methods and details on case data at https://coronavirus.data.gov.uk/details/about-data; an estimated 27% of COVID-19 infections were detected during the study period[16]). Using these data, we evaluate the effectiveness of policies in reducing international importations and how they contributed to the establishment and local transmission dynamics of Delta in England. We then investigate, at a high spatial resolution, how human mobility contributed to context-specific growth of Delta in England.

## International importations of Delta

To provide a global context for the emergence of Delta in the UK, we first conducted a phylodynamic analysis by uniformly subsampling SARS-CoV-2 Delta genome sequences by collection date between 4 March 2021 and 15 June 2021 (*n* = 975). Details of the origin and spread of Delta within India are uncertain. A substantial increase in genomic surveillance across the country would probably facilitate the study of the emergence and expansion of Delta there, but is outside the scope of this work. To put the UK epidemic into context, we estimated the time of the most recent common ancestor (TMRCA) of Delta globally to be 19 October 2020 (95% highest posterior density (HPD) interval: 6 September 2020 to 29 November 2020). The relative frequency of Delta in India does not appear to increase substantially until March 2021 (Fig. 1), coinciding with a rapid expansion in case numbers there (Extended Data Fig. 1) and a decline in the relative frequency of genomes assigned to Kappa, a sibling lineage of Delta (https://www.gisaid.org/). Genomic surveillance in India revealed that several sub-lineages of Delta existed prior to its expansion in March[17] (Fig. 1a,b). This standing diversity is consistent with undetected transmission of Delta in India between late 2020 and March 2021.

We evaluated the global dissemination of Delta from March 2021 by multiplying, for each country, estimated numbers of SARS-CoV-2 cases, relative frequencies of Delta, and relative numbers of outward international passengers (estimated exportation intensity (EEI); Methods). The EEI of Delta increased rapidly during March 2021 and was highest around late April 2021, coinciding with its peak incidence in India (Extended Data Fig. 1). The subsequent rapid spread of Delta in the USA, Russia, UK, Mexico and elsewhere and its decline in India resulted in the former locations becoming main exporters of Delta by June 2021 (Extended Data Fig. 1), corroborating global trends in Delta phylogeography (Fig. 1a) and reported cases (Fig. 1b). Similar patterns of rapidly changing foci of international dissemination were observed for the initial wave of SARS-CoV-2 in 2020[18].

To evaluate the temporal dynamics of Delta importation into England and to reconstruct its subsequent local spread, we conducted a travel history-aware Bayesian phylogeographic analysis[19] of 93,649 Delta sequences, from GISAID and COVID-19 Genomics UK Consortium (COG-UK), which accounts in part for the phylogenetic uncertainty inherent in SARS-CoV-2 phylogenies[18]. To render the analysis tractable, we split the full tree into three independent subtrees (Fig. 1a) prior to phylogeographic analysis. Virus genomes were generated from around 40–60% of all positive cases in England during the emergence of Delta between March and May 2021[20] (Fig. 1c) and combined with metadata on the locations (at the upper tier local authority (UTLA) level), enabling us to trace the introduction of the virus and characterize its spread at a high spatio-temporal resolution[20].

We estimated a minimum of 1,458 (95% HPD 1,398–1,513) separate international introductions of Delta into England, with approximately half inferred to have originated from India (posterior mean 56.5%; 95% HPD 53.7%–59.1%). We found that the majority of Delta genomes in England can be traced back to introductions that are inferred to have occurred prior to the implementation of a mandatory hotel quarantine

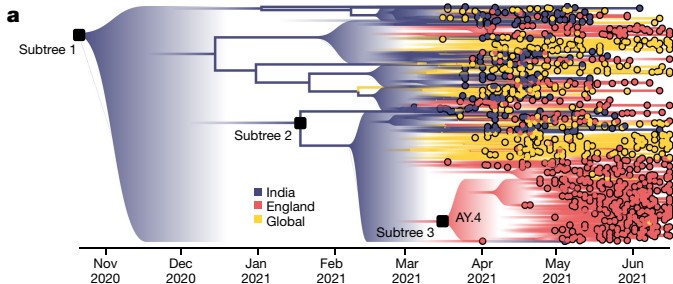

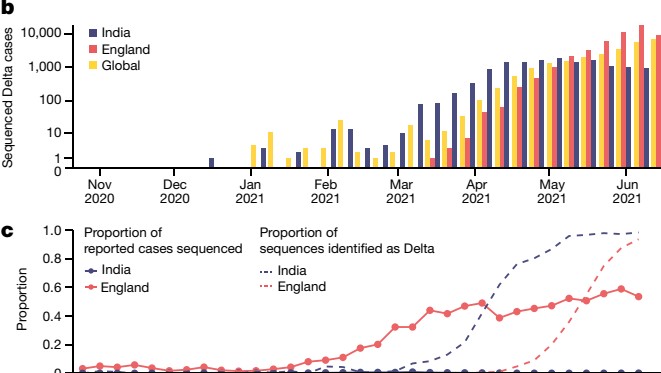

**Fig. 1 | The emergence and rapid geographic expansion of Delta.**
**a**, Time-calibrated phylogenetic reconstruction of Delta based on 1,000 sequences subsampled from 93,649 sequences from 100 countries (52,992 from England). The tree was split into three subtrees (with *n* = 28,783, 28,715 and 36,151 sequences, respectively) prior to full analysis. The roots of these three subtrees, and of lineage AY.4 are labelled with black squares. Lineage colours represent the inferred countries and/or regions where transmission occurred. **b**, The number of sequenced cases of Delta per week in India, England and globally, where 'global' refers all countries other than England and India. **c**, The proportion of sequenced, reported positive cases in India and England (solid lines, *n* = 52,992 sequences from England, corresponding to 84% of all sequences from the UK during the study period) and the proportion of sequenced cases classified as Delta in India and England (dashed lines).

for people arriving from India on 23 April 2021 (posterior mean 84.3%; 95% HPD 77.8–90.4%). During this period, 90.0% of introductions are inferred to have originated from India (95% HPD 86.5–93.1%). These inferred importation dynamics closely match data on individual travel histories obtained from infected incoming international passengers (origin–destination travel histories are available for 1.4% of genomes; *n* = 770) (Fig. 2b and Extended Data Fig. 2).

The high variation in sampling intensity among countries (specifically, the higher sampling intensity in England than in other countries) means that the true number of importations into England is probably much larger than that inferred from phylogeographic analysis alone (Fig. 1b,c; see the related discussion in the context of the first wave in the UK[18]). For example, the AY.4 lineage (Fig. 1a) comprises 42,445 sequences and was probably imported to England many times. We investigated AY.4 by pairing genomic data with contact tracing data collated by Public Health England (now the UK Health Security Agency). During the study period we found 61 sequenced cases with AY.4 had a travel history from India and 140 had a travel history from elsewhere, similar to the time-varying importation dynamics seen across the entire dataset (Fig. 2a and Extended Data Fig. 2). Thus, sampling heterogeneity means that the number of importations estimated from phylogenetic analysis represents a lower bound on the true number[18].

To investigate the importation of Delta into England specifically, and to cross-validate the results above using independent data, we use the estimated importation intensity (EII) of Delta to England over time[18,21].

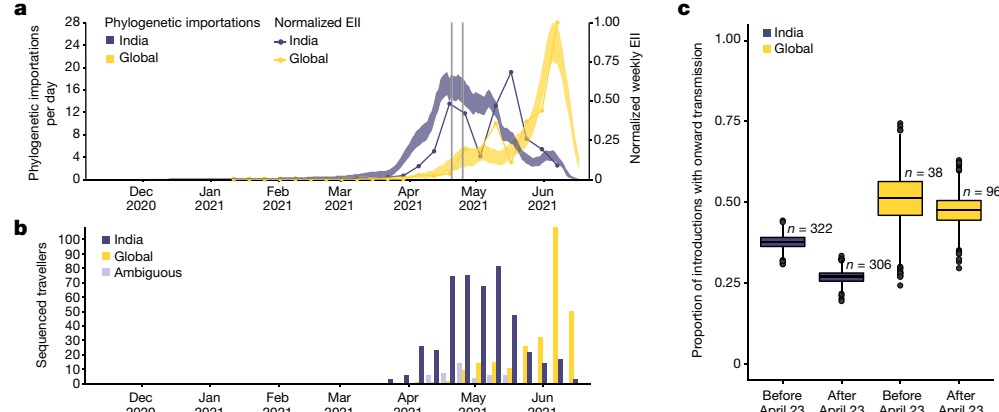

**Fig. 2 | Timing of importations of Delta into England. a**, The estimated daily number of importations of Delta from India (blue shaded area) and other countries (yellow shaded area), inferred from phylogenetic analysis. Shaded areas show 95% HPDs of the estimate. Blue and yellow lines show the EII of Delta, obtained by combining data on human movements, cases and prevalence of Delta, normalized to the same scale as the phylogenetic estimates. Grey vertical lines show the timing of the announcement of travel restrictions from India to England (18 April 2021) and their implementation on 23 April 2021. **b**, Temporal distribution of genome sequences from cases with a known travel history from India (blue) and other countries (yellow). Isolates with recent travel to both India and other countries are considered ambiguous (grey). **c**, The proportion of all virus introductions that show evidence of onward transmission in the UK, estimated separately for weeks before and after the implementation of hotel quarantine (23 April 2021) and stratified by the location of origin (India, blue; other countries, yellow). The box plot displays the median, with lower and upper hinges representing the 25th and 75th percentiles of each group. Whiskers extend to the most extreme data points no more than 1.5 times the interquartile range beyond each hinge. The number of observations in each group is annotated above each box.

The EII is a metric of Delta importation that represents trends in the number of Delta cases arriving in the country, irrespective of whether or not those cases result in local transmission. This is distinct from the phylogenetic analysis above, which better captures trends in the number of Delta introductions that did lead to forward transmission in England. The EII combines (1) weekly reported cases, (2) weekly prevalence of Delta genomes, and (3) weekly aggregate human mobility (inferred from mobile phone data) into England through direct connections (Fig. 2a; see refs. [18,21] for related approaches). The EII from India increased rapidly in April 2021 following the rise in cases in India, and remained high until the end of May 2021. EII correlates strongly with the inferred importations from genomic data but is weaker for imports from India after the implementation of hotel quarantine (Extended Data Fig. 3). We then estimate, from genomic data, the proportion of inferred importations that led to observed onward transmission in the community (defined as at least one ancestral node in England), stratified by location of origin in the three weeks pre- and post-implementation of hotel quarantine. We find that pre-quarantine, 37.7% (95% HPD 34.0%–41.7%) of importations from India led to observable onward transmission. After the implementation, the fraction of importations from India leading to observed onward transmission dropped to 26.9% (95% HPD 23.0%–30.2%) (Fig. 2c). For comparison, the proportion of introductions from other locations leading to onward transmission did not change during the two periods (around 50%) (Fig. 2c). The decrease in onward transmission is most apparent in importations associated with travel history, which suggest the trend is driven by the implementation of hotel quarantine and not temporal biases in lineage detection (Extended Data Fig. 3). Even though we observe that the implementation of hotel quarantine was effective in reducing onward transmission, substantial importation had already occurred before its implementation and additional introductions from other countries probably further accelerated the spread of Delta in England from May onwards (Fig. 2a).

There are several reasons why some importations led to onward transmission within England after the implementation of hotel quarantine for arriving travellers: (1) a separate terminal for arrivals from mandatory quarantine countries was not opened at the UK's largest airport (London Heathrow) until 1 June 2021[22], so arriving passengers may have mixed with others before entering mandatory quarantine; (2) individuals may have become infectious and transmitted the virus only after leaving quarantine, either owing to an unusually long latent period or within-group transmission during the quarantine period; (3) individuals may have infected others on a connecting flight where the connecting airport did not require hotel quarantine; (4) there were exemptions to hotel quarantine that may have led to onward transmission in the community[23].

## Lineage dynamics of Delta in England

Importations of Delta occurred on a background of relaxation of social distancing in England: on 12 April 2021, outdoor dining and non-essential retail reopened, and on 17 May 2021, restrictions on indoor dining and international travel were relaxed[24]. The relative frequency of Delta genomes in England increased rapidly during May and the number of reported COVID-19 cases subsequently increased[25] (Fig. 1c). Initially, Delta transmission clusters were concentrated in the North West region of England and were commonly associated with returning travellers[26]. We sought to reconstruct the dispersal dynamics of independently imported Delta transmission lineages within England, in the context of changing NPIs.

We analysed all identified Delta transmission lineages in England and inferred their history of dissemination among subnational regions (UTLAs). Sequence sampling was highly representative of reported cases at the UTLA level (Extended Data Fig. 4), making possible the reconstruction of virus movements across England using continuous phylogeography approaches[27]. We observe high heterogeneity among UTLAs in the numbers of Delta introductions from other English regions (Fig. 3a), with Lancashire and Greater Manchester each receiving more than 2,000 estimated independent introductions and Torbay receiving only 9. The majority ($n$ = 11,960) of Delta sequences in England belong to a single transmission lineage (lineage I) (Fig. 3d), which was sampled mostly in Greater Manchester and Lancashire, and we observe many short-range lineage movements among UTLAs in these areas (Fig. 3a). Greater London also received many Delta cases from elsewhere in England (Fig. 3a), as expected, given its population size and connectedness to other metropolitan areas[21]. Transmission lineages II and III each comprised 3,000–4,000 genomes; lineage II is

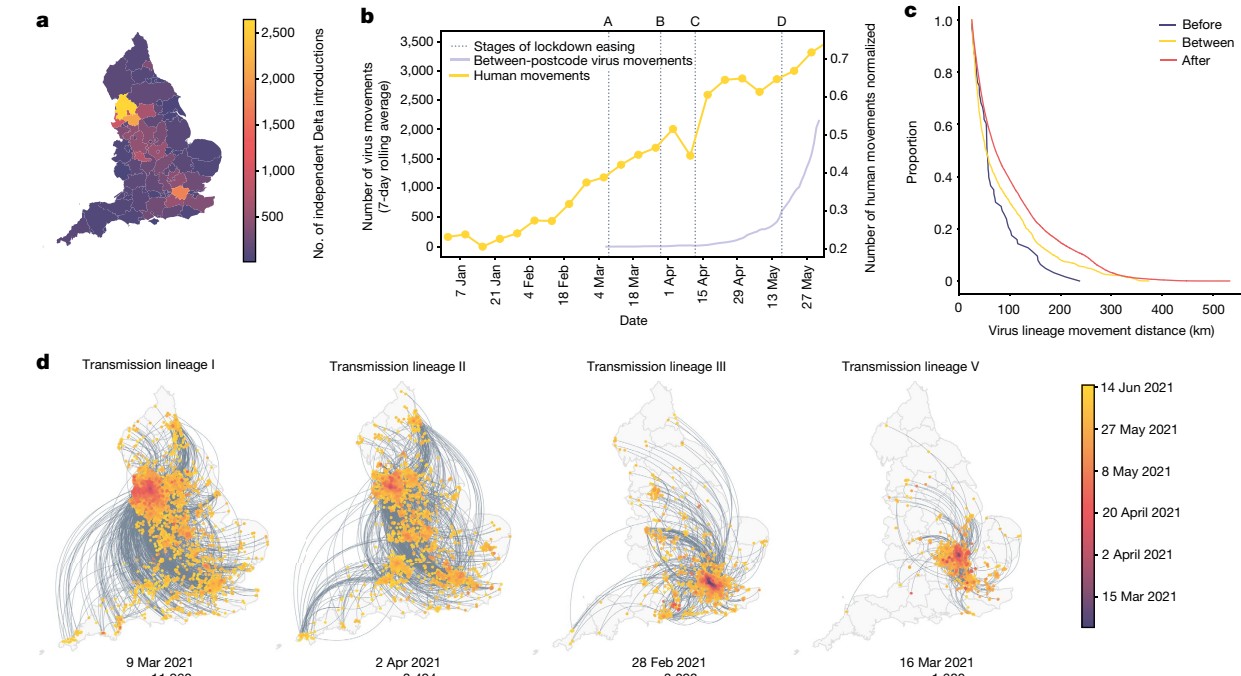

**Fig. 3 | Introductions and regional dynamics of Delta transmission lineages. a**, The number of independent introductions per UTLA in England derived from continuous phylogeographic analysis of all Delta transmission lineages with more than five sequences. **b**, Trends in aggregate intra- and inter-UTLA mobility normalized to pre-pandemic levels (yellow) and the number of virus lineage movements between postcode districts. Letters denote stages of lockdown easing: A, schools re-open and limited outdoor mixing between households is permitted (8 March 2021); B, 'stay at home' directive is lifted, more outdoor mixing (up to six people from two households) is allowed (29 March 2021); C, non-essential retail, holiday lets and campsites re-open and outdoor dining is permitted (12 April 2021); D, indoor hospitality

re-opens and indoor mixing is permitted (17 May 2021). **c**, The proportion of virus lineage movements between postcodes more than 25 km apart: the *y* axis denotes the proportion of movements that are less than or equal to the value on the *x* axis. This is shown for movements before lockdown easing (C (blue)), between C and D (yellow) and D (red). C and D are defined in **b**. **d**, Virus lineage movements inferred by continuous phylogeographic analysis for four example large transmission lineages (transmission lineages IV, VI and VII are shown in Extended Data Fig. 5). The direction of lineage movement is anticlockwise, and dots represent the start and end points of movements, coloured by inferred date. The size of each lineage and its inferred TMRCA date are shown below each map. Distance kernels for each lineage are shown in Extended Data Fig. 7.

distributed across multiple urban areas (especially in the North West), whereas the latter is focused in Greater London and the South East (Fig. 3d). We also highlight transmission lineage V (Fig. 3d), originally centred in Bedfordshire, the location of one of the first Delta outbreaks in England that was subject to surge testing[28] (Extended Data Fig. 5).

In early May 2021, the number of virus lineage movements among locations accelerated (Fig. 3b and Extended Data Fig. 5), showing that increase in Delta frequency (Fig. 1c) was associated with regional dissemination. This spread occurred on the background of relaxing NPIs and increased mixing (between mid-January and June 2021, mobility in England increased from 20% to 70% of its pre-pandemic level, and estimated mean daily contacts increased[29] from approximately 2 to 5). By contrast, the initial wave of SARS-CoV-2 introductions to the UK in spring 2020 occurred during a period of increasing travel and social restrictions[18]. In general, we find that as NPIs were progressively relaxed over time, long-range viral lineage movements comprised an increasing proportion of all movements (Fig. 3c).

For the seven largest Delta transmission lineages in England (I–VII) we observed approximately three times more exports from Greater Manchester than from Greater London. Further, we observe that Bolton, Blackburn, Salford, Bury and Greater Manchester had on average higher than expected numbers of exportations for their population sizes (Extended Data Fig. 6). This difference matches early epidemiological data: the largest and earliest Delta outbreaks were in North West England (on 21 May, Bolton had 452 cases per 100,000 population and Greater London had 21.6 cases per 100,000 population) (https://coronavirus.data.gov.uk/; Methods). Introductions of Delta into other, smaller urban areas also spread rapidly (for example, transmission

lineage V) (Fig. 3d) and were important for the propagation of the variant across England. We observe a spatial structure of the seven largest lineages; the frequency of viral movements declined with the distance away from the origin location but we also observe a second peak at around 260 km (similar to the distance between Greater London and Greater Manchester) (Extended Data Fig. 7). Although North West England was a focus of early Delta transmission, the Delta epidemic in England derived from many successful independent international importations. Each of the main Delta transmission lineages in England grew at a similar rate (Extended Data Fig. 8). By contrast, the Alpha variant expanded across the UK from a single origin in South East England[21]. The spatial expansion of Delta transmission lineages plateaued after early June, when most UTLAs had established Delta transmission and the relative frequency of Delta genomes in England had exceeded 90% (https://covid19.sanger.ac.uk/lineages/raw).

Although Scotland, Wales or Northern Ireland were not included here, case count data suggest that cities in England (https://coronavirus.data.gov.uk/details/download) were the main source of the expanding Delta epidemic in the UK; owing to this source-sink structure we do not anticipate that omitting these countries substantially affects our reconstruction of epidemic dynamics in England (of the Delta genomes available before 15 June 2021, 57,592 were from England, 9,738 were from Scotland, 1,067 were from Wales and 325 were from Northern Ireland).

## Factors contributing to the growth of Delta

Regional and international heterogeneity in incidence, vaccination and human mobility determine the dynamics of infectious diseases[30],

including those of SARS-CoV-2[18,31,32]. We used a combination of epidemiological, aggregate human mobility and genomic data to test whether relaxation of NPIs, virus importations and vaccination rates correlate with local Delta growth rates. To do so, we developed a hierarchical Bayesian model to estimate the effect of these factors on the weekly relative growth of Delta (that is, the weekly change in the observed proportion of Delta genomes on a log odds scale[33]) at the UTLA level for England. Models for estimating the increase in transmissibility of new variants are typically based on increases in relative frequency[1,33] but rarely take into account other potential confounding factors, such as variation in population behaviour, vaccination rates or numbers of independent virus introductions.

In general, growth rates varied widely across locations and weeks in England (Fig. 4a and Extended Data Fig. 10). Our model estimates that the most important tested predictor of the variation in growth of Delta (relative to Alpha) across UTLAs in England was within-UTLA mixing (that is, relative changes in weekly within-UTLA human mobility, compared with the pre-pandemic period) (Fig. 4a,b and Supplementary Tables 5 and 6). The importance of within-UTLA mobility as a factor during the emergence of a new variant (until Delta prevalence reached 85% (25%–75% quantiles: 78%–96%)) is unsurprising, as pre-emptive restrictions on movement and social mixing slow the emergence of new pathogens or variants[34] (see counterfactual scenarios in Extended Data Fig. 10); the cost/benefit ratio of such restrictions will of course depend on the specific context of variant emergence. The relaxation of NPIs therefore increased both within- and among-region transmission (see Fig. 3c). Other European countries did not observe such a rapid increase in Delta relative frequency during May 2021 (https://www.gisaid.org/); possible reasons for this difference are (1) during this period, levels of mobility and mixing (both local and regional) were lower in those countries and/or (2) those countries potentially received fewer international importations of Delta (86,489 passengers flew from India to the UK between March and June, whereas 43,515 flew to Germany, and 16,688 flew to France, during the same period).

Model fit did not improve when including weekly numbers of independent viral introductions estimated from genomic data or vaccination rates (Supplementary Table 6). We refrained from translating estimates of the growth rate of Delta relative frequency into differences in the reproduction number, as this is sensitive to assumptions about the generation time of the variant, which is also influenced by NPIs and immunity[35]. Further studies should consider estimating the generation times of VOCs in specific contexts of immunity, NPIs and household structure[36] to accurately translate relative growth rates into $R_t$.

## Discussion, limitations and future work

We find that growing epidemics of SARS-CoV-2 Delta worldwide led to a wave of importations of the VOC into England, initially from India, and later from other countries. These importations found fertile ground as they arrived in a context of easing social restrictions, and consequently expanded rapidly across England. Much transmission occurred in unvaccinated and younger populations (https://coronavirus.data.gov.uk/details/download), and high levels of Delta transmission within the UK led to onward dissemination of the variant to other countries (see for example, ref. [37]). By pairing the phylogenetic results with contact tracing data we conclude that hotel quarantine measures were effective in reducing onward transmission of imported Delta cases in England. However, after 21 May 2021, we found that levels of local social mixing in England—and not numbers of importations—were associated with faster relative growth of Delta. At that point, the independently introduced transmission lineages grew at a similar pace; details of their geographic distribution and expansion will support future work defining the optimal spatial interventions to reduce transmission of VOCs in England.

Compared with the Alpha variant, which arose and spread from a single location in South East England[18], the expansion of the Delta

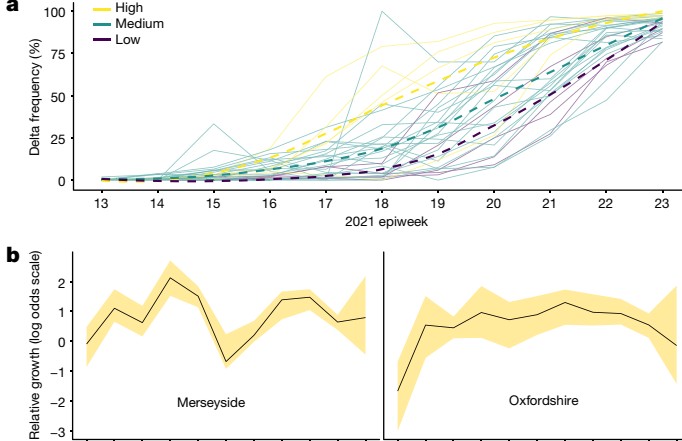

**Fig. 4 | Variation in Delta growth rates across UTLAs in England. a**, The increase in Delta frequency compared with Alpha at the UTLA level. UTLAs are coloured according to the level of average within-UTLA mobility: high, the five UTLAs with the highest within-location mobility; low, the five UTLAs with the lowest within-location mobility; medium, the remaining UTLAs. Solid lines show data for given UTLAs; dashed lines show LOESS curves fit to the data for each mobility category. **b**, Examples of weekly growth (the solid line corresponds to posterior medians) of UTLAs with high (left) and low (right) within-UTLA mobility. The shaded regions represent the corresponding 95% Bayesian credible intervals (2.5th and 97.5th quantiles of the posterior distribution). In **a**, data are shown only for UTLAs with 500 or more sequenced samples.

variant was predominantly owing to exports from the North West (Fig. 3). Analysis of the Alpha variant and of the first wave of SARS-CoV-2 in spring 2020 suggested that Greater London had a substantial role in spreading SARS-CoV-2 across England, as expected, given it is the largest city in England by far, and is highly connected by road, rail and air to other locations. However, Greater London was less important in the spread of Delta, even after Delta had become established there. This indicates the importance of founder effects; where a VOC first becomes established within a country may have a strong effect on subnational spatial dissemination, and this information is useful for planning localized interventions.

Furthermore, although there are intrinsic differences in transmissibility between VOCs, the role of NPIs and levels of immunity from prior infection or vaccination also affect their dynamics. After the start of the first UK national lockdown during the first wave of infections in 2020, lineage movements were severely curtailed and most lineages went extinct[18]; by contrast, the viral movement of Delta lineages increased after the relaxation of NPIs, accompanied with a subsequent rise in positive cases (Fig. 3). For the most recent VOC, Omicron, NPIs have remained relatively stable throughout England, and the increase in cases of the Omicron subvariant BA.2 in more rural areas in the South West in February and March 2022 has been speculated to be a result of lower infection rates there during the previous Omicron BA.1 wave[38] (December 2021–January 2022). Therefore, the effect of seeding location, immunity from previous waves or vaccination and NPI changes all contribute to the large and continued spatial heterogeneity in the spread of VOCs.

The undetected genetic diversity and uneven sampling of Delta in India make the precise estimation of the number of importations to England difficult to achieve from genetic data alone[27] (Extended Data Fig. 9). However, our phylogenetic estimates correlate strongly with estimates derived from independent data on case incidence, Delta prevalence and arriving travellers (EII) (Methods and Fig. 2c) during the period before quarantine policies were announced. Fortunately,

additional contact tracing data from public health agencies enabled us to overcome the limitations inherent in the unevenly sampled global virus genomic dataset and provide additional confidence in our findings.

Our statistical analysis shows that higher Delta growth rates were positively associated with levels of local mixing in England. The existence and magnitude of future NPIs needed to reduce the healthcare burden of future VOCs to sustainable levels will depend on the local levels of population immunity (from vaccination and prior infection). Future work should focus on identifying the factors that are most conducive to spread in particular contexts (for example, high versus low NPI regimes and across levels of population immunity[39]) so that responses can be planned accordingly. This will require a better characterization of the distribution and variation of infectiousness over time, and an understanding of the virus generation time in different behavioural contexts[40]—for example, among individuals who are vaccinated or unvaccinated and/or those who have had previous exposure to SARS-CoV-2 (including knowledge of the lineage or variant). To do so effectively will require investments in large-scale and coordinated serological studies[41], especially for VOCs with the ability to evade immunity.

Even though reporting of case numbers, virus genomic surveillance, sampling strategies and mobile phone penetration differ across the world, our estimates can still provide qualitative insights into the trends in the source locations and the rates of international importation. Including estimates of probable importations in disease surveillance programmes may help support public health decision making[42], and further improvements in these estimates can be achieved when global health surveillance systems are more integrated, and investments in data generation and capacity are linked directly to paired genomic–epidemiological analytical pipelines.

The detail with which we document the spatial invasion process of Delta in England provides an opportunity to re-examine how more spatially targeted interventions can support COVID-19 control in the future. Our work highlights the relative importance of local (within country) behavioural and mobility changes in determining the speed at which Delta spread in England; such changes will probably be more important than international travel restrictions during the emergence of future variants. Globally coordinated data and analytical pipelines that capture heterogeneity in virus circulation, immunity and policy responses will be necessary to produce the insights necessary to curb the spread of emerging infectious diseases and new variants. However, they can only be successful when integrated into a public health framework that can respond and adapt rapidly to public health threats during their emergence.

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

[1]Institute of Evolutionary Biology, University of Edinburgh, Edinburgh, UK. [2]Department of Zoology, University of Oxford, Oxford, UK. [3]College of Engineering, Mathematics and Physical Sciences, University of Exeter, Exeter, UK. [4]MRC Centre of Global Infectious Disease Analysis, Jameel Institute for Disease and Emergency Analytics, Imperial College London, London, UK. [5]Section of Epidemiology, Department of Public Health, University of Copenhagen, Copenhagen, Denmark. [6]Molecular Immunity Unit, Department of Medicine, Cambridge University, Cambridge, UK. [7]Spatial Epidemiology Lab (SpELL), Université Libre de Bruxelles, Bruxelles, Belgium. [8]Department of Microbiology, Immunology and Transplantation, Rega Institute, KU Leuven, Leuven, Belgium. [9]Google, Mountain View, CA, USA. [10]Department of Mathematics, School of Science and Engineering, Tulane University, New Orleans, LA, USA. [11]Department of Infectious Disease, Imperial College London, London, UK. [12]UK Health Security Agency, London, UK. [13]Instituto de Medicina Tropical, Faculdade de Medicina da Universidade de Sao Paulo, Sao Paulo, Brazil. [14]BlueDot, Toronto, Ontario, Canada. [15]Divisions of Internal Medicine and Infectious Diseases, Toronto General Hospital, University Health Network, Toronto, Ontario, Canada. [16]Department of Medicine, Division of Infectious Diseases, University of Toronto, Toronto, Ontario, Canada. [17]Li Ka Shing Knowledge Institute, St Michael's Hospital, Toronto, Ontario, Canada. [18]Department of Biosystems Science and Engineering, ETH Zurich, Zurich, Switzerland. [19]Swiss Institute of Bioinformatics, Lausanne, Switzerland. [20]Wellcome Sanger Institute, Wellcome Genome Campus, Hinxton, UK. [21]Centre for Genomic Pathogen Surveillance, Big Data Institute, Nuffield Department of Medicine, University of Oxford, Oxford, UK. [22]Pathogen Genomics Unit, Public Health Wales NHS Trust, Cardiff, UK. [23]School of Biosciences, The Sir Martin Evans Building, Cardiff University, Cardiff, UK. [24]Quadram Institute, Norwich, UK. [25]Institute of Microbiology and Infection, University of Birmingham, Birmingham, UK. [26]Departments of Biostatistics, Biomathematics and Human Genetics, University of California, Los Angeles, Los Angeles, CA, USA. [27]Department of Pathobiology and Population Sciences, Royal Veterinary College London, London, UK. [28]Pandemic Sciences Institute, University of Oxford, Oxford, UK. [29]These authors contributed equally: John T. McCrone, Verity Hill, Sumali Bajaj, Rosario Evans Pena. [30]These authors jointly supervised this work: Oliver G. Pybus, Andrew Rambaut, Moritz U. G. Kraemer. *A full list of members and their affiliations appears in the Supplementary Information. ✉e-mail: opybus@rvc.ac.uk; a.rambaut@ed.ac.uk; moritz.kraemer@biology.ox.ac.uk

## Methods

### Genomic data

International (non-UK) sequences were downloaded from GISAID on 15 September 2021 and combined with sequences from England taken as part of community surveillance (pillar 2) available from COG-UK as of September 2021. Each week, each of the pillar 2 testing laboratories selected a number of 96-well plates proportional to the fraction of all testing done at the laboratory, for sequencing. Even though the instruction was that these should be selected randomly, we cannot exclude the possibility of some level of error. However, at the scale at which the COG-UK consortium is operating, we do not anticipate that this affects the results of the study. Sequences were processed and aligned as part of the daily datapipe analysis managed by CLIMB on behalf of COG-UK. Duplicate and environmental sequences, as well as those with impossible or incomplete collection dates, were removed. All sequences were aligned to the reference Wuhan-Hu-1 (GenBank accession MN908947.3) with minimap2 and samples with less than 93% coverage were discarded. Scorpio (https://github.com/cov-lineages/scorpio) was run as part of Pangolin[43], and sequences containing the Delta VOC constellation of mutations were kept for further analysis.

Problematic sites in the resulting alignment were masked prior to phylogenetic inference and isolates with known sequence artefacts were removed (see https://github.com/COG-UK/Delta-analysis for details). Additionally, mutations in the Delta VOC have caused widespread amplicon dropout of amplicon 72 in the commonly used ARTIC primer scheme (https://www.protocols.io/view/ncov-2019-sequencing-protocol-v3-locost-bh42j8ye) before the introduction of version 4 of the primer scheme. To avoid spurious phylogenetic associations based on differential treatment of amplicon dropout with COG-UK and across the globe, we masked sites 2142–21990, which represent the region solely covered by amplicon 72 and are not overlapped by neighbouring amplicons. Delta sequences from India were highly heterogeneous in space (Extended Data Fig. 9).

### Phylogenetic analyses

To provide an overview of the global expansion of Delta (Fig. 1a), we analysed a subset of 1,000 Delta genomes sampled evenly through time. To minimize the effect of incorrectly reported collection dates, we restricted our analysis to samples where the lag between sample collection date and GISAID submission date is less than four weeks. To further ensure only the highest quality samples were included, we built an maximum likelihood tree using iqtree2[44], rooted with Wuhan-Hu-1 (GenBank accession MN908947.3) as an outgroup, and used Treetime[45] to remove tips lying beyond two interquartile ranges from the regression of time against root-to-tip distance. This analysis resulted in a final dataset of 975 samples. The temporal tree estimated by treetime was used as a starting tree in the following Bayesian analysis with slight modifications to randomly resolve polytomies. Two chains of 100 million states were run using BEAST v1.10.4[46] with sampling every 20,000 states. Both chains were combined with the first 10 million states removed for burnin. We used a HKY + $\Gamma$ substitution model[47], a flexible Skygrid coalescent prior[48] with grid points every two weeks[45], and an asymmetric, discrete phylogeographic model with samples assigned to Indian, English and global locales. Preliminary analysis showed very little temporal signal in the data, which is unsurprising given the relatively slow evolutionary rate of SARS-CoV-2 and the short study period. Therefore, in all analyses the evolutionary rate was fixed to $7.5 \times 10^{-4}$ substitutions per site, as estimated in ref. [18]. Convergence was assessed using Tracer v1.7[49].

The goal of our phylogenetic analysis was to accurately and efficiently describe importation dynamics into England, without sacrificing the dense sampling needed to reconstruct internal spread at a high resolution. Owing to the large size of the required dataset, we followed a similar phylogenetic approach to that used in ref. [18]. First, an approximately maximum likelihood phylogeny was built using a JC69 substitution model in FastTree[50], and rooted on Wuhan-Hu-1 (GenBank accession MN908947.3), a high quality Pango lineage B sample from 2019-12-26, as an outgroup. Internal branches representing less than one substitution were collapsed to polytomies. This tree was then split into three subtrees of roughly equal size (Fig. 1a) (28,783, 28,715 and 36,151 tips). As above, Treetime[45] was then used to remove temporal outliers, generate a starting time tree, and estimate the number of mutations along each branch. For subtree an empirical distribution of time trees was estimated independently using a recently implemented model in BEAST v1.10[46] (commit:d1a45) which replaces the substitution model in classical analyses. In brief, in this approach the likelihood of the number of mutations along each branch was calculated from a Poisson distribution with mean equal to the evolutionary rate multiplied by the length of the branch in time[51]. In this approach, the standard topological tree search is constrained to operators that sample node heights and resolutions of polytomies present in the substitution tree.

For each subtree, 50 MCMC chains of 40 million iterations were run, sampling trees every 2 million states with the first 20 million states removed as burnin, resulting in datasets of 514–520 empirical trees. The analyses were run using a flexible Skygrid coalescent prior[48] with grid points every two weeks[45]. Model convergence and proper statistical mixing were verified in Tracer v1.7[49].

The empirical trees sets estimated above were used to reconstruct importations into England under an asymmetric discrete phylogeographic model. Taxa were split into three locations: England, India and global, with the global state representing all countries other than England and India. We used the recently developed travel-aware phylogenetic model available in BEAST v1.10[19] to better inform the transition rates in the reconstructed phylogeography. 'Travel history' nodes were placed 1 week before isolates from England with known travel history. Where such travel included both India and other countries, ambiguous non-UK states were used. We ran eight chains of 625,000 states, sampling every 2,250 states and with the first 62,500 states removed as burnin, resulting in a total of 1,998 trees sampled from the posterior distribution. Introductions were defined as nodes inferred to be in England with parents in either India or the catch-all global location. The date of importation was assumed to be half-way between such a node and its parent.

Following the importation analysis, the seven largest importations (those with >1,500 sequences, $n = 25,983$) were selected, as well as all importations with five or more sequences, from a representative tree from the posterior set with the same number of total importations as the posterior median. Within this analysis, only sequences with unambiguous postcode districts were used, resulting in a dataset of 25,139 sequences for the seven largest transmission lineages and 24,411 across 280 smaller lineages, which were extracted from the master COG-UK alignment, described in 'Genomic data' above. Within those postcode districts, we assigned random coordinates to each sequence, as the continuous phylogeographic analysis does not permit identical values. This was achieved using geographical data from[52]. We then reconstructed the geographic movement of nodes on a fixed tree (pruned from the overall maximum clade credibility (MCC) tree) in BEAST v.1.10[46], using a relaxed random walk model[53], and a Cauchy distribution to account for among-branch heterogeneity in dispersal velocity. Large lineages were inferred independently, and all small lineages were inferred in a single run, with the shared parameters for likelihood, precision, and covariance of coordinates, but independent estimates of diffusion rate and trait likelihood. Following this run, 22 small introductions were removed due to their chains not converging to the same posterior. An MCC tree was then generated using TreeAnnotator[46] to summarize the posterior tree distribution for all lineages. Visualizations were made using a custom Python script. XML files were generated using beastgen.py (https://github.com/ViralVerity/beastgenpy) and can be found along with data processing and visualization scripts on GitHub.

For the export analyses we compare Greater London to Greater Manchester which consists of the UTLAs Salford, Trafford, Stockport, Oldham, Bolton, Tameside, Bury, Rochdale, Wigan and Manchester.

**State-level incidence data from India.** State-level COVID-19 case count data were extracted from https://api.covid19india.org/csv/latest/states.csv.

**Incidence data from England.** COVID-19 case count data for each Local Tier Local Authority were downloaded via https://coronavirus.data.gov.uk/details/download.

### Travel history data

Four sources of data were compiled to provide the travel history for laboratory-confirmed cases, depending on availability for each individual case: (1) public health passenger locator forms are required for entry into the UK; (2) routine public health contact tracing data including UK Health Security Agency Second Generation Surveillance System (SGSS)[54], (3) COVID-19 test requests with reported travel associations and (4) responses to additional telephone interviews for cases.

### Covariate processing for statistical analyses

**COVID-19 case count and vaccination data for the UK.** COVID-19 case count data and cumulative vaccination data were downloaded by UTLA from 30 January 2020 to 28 July 2021 by specimen and dosage date, respectively, via https://coronavirus.data.gov.uk/details/download. These data include positive laboratory-based PCR tests and positive lateral flow tests, but do not include tests where the lateral flow test was positive and PCR follow-up tests were negative (further details at https://coronavirus.data.gov.uk/details/about-data). The COVID-19 case count at the UK country level was calculated by aggregating case data on the UTLA level. Additionally, to match the genomic data, the COVID-19 case count and vaccination data for some UTLAs were aggregated under an area code made up of these multiple UTLAs (see Supplementary Table 3). All entries with the recently discontinued area code E10000002 were assigned the new area code E06000060.

**UK population data.** UTLA-level 2020-mid-year population size estimates were downloaded from https://www.ons.gov.uk/peoplepopulationandcommunity/populationandmigration/populationestimates/datasets/populationestimatesforukenglandandwalesscotlandandnorthernireland. Population size data were used to calculate the proportion of the population that was partially or fully vaccinated in a location.

**Global population data.** Country-level population size estimates for the year 2021 were downloaded from https://data.worldbank.org/indicator/SP.POP.TOTL?name_desc=false.

**Aggregated and anonymised human mobility data.** We used the Google COVID-19 Aggregated Mobility Research Dataset[31,55], which contains anonymized relative mobility flows aggregated over users who have turned on the 'location history' setting, which is turned off by default. This is similar to the data used to show how busy certain types of places are in Google Maps, helping identify when a local business tends to be the most crowded. The mobility flux is aggregated per week, between pairs of approximately 5 km² cells worldwide, and for the purpose of this study further aggregated for LTLAs in the UK (https://geoportal.statistics.gov.uk/datasets/lower-tier-local-authority-to-upper-tier-local-authority-december-2016-lookup-in-england-and-wales/explore) and to the country level (https://gadm.org/) for all other countries for the time period of 29 October 2020 to 6 June 2021.

To produce this dataset, machine learning is applied to log data to automatically segment it into semantic trips. To provide strong privacy guarantees[56], all trips were anonymized and aggregated using a differentially private mechanism to aggregate flows over time (see https://policies.google.com/technologies/anonymization). This research is done on the resulting heavily aggregated and differentially private data. No individual user data was ever manually inspected; only heavily aggregated flows of large populations were handled. All anonymized trips are processed in aggregate to extract their origin and destination location and time. For example, if users travelled from location $a$ to location $b$ within time interval $t$, the corresponding cell $(a,b,t)$ in the tensor would be $n \pm err$, where err is Laplacian noise. The automated Laplace mechanism adds random noise drawn from a zero-mean Laplacian distribution and yields $(\epsilon,\delta)$-differential privacy guarantee of $\epsilon = 0.66$ and $\delta = 2.1 \times 10^{-29}$ per metric. Specifically, for each week $W$ and each location pair $(A,B)$, we compute the number of unique users who took a trip from location $A$ to location $B$ during week $W$. To each of these metrics, we add Laplace noise from a zero-mean distribution of scale $1/0.66$. We then remove all metrics for which the noisy number of users is lower than 100, following the process described[56], and publish the rest. This yields that each metric we publish satisfies $(\epsilon,\delta)$-differential privacy with values defined above. The parameter $\epsilon$ controls the noise intensity in terms of its variance, while $\delta$ represents the deviation from pure $\epsilon$-privacy. The closer they are to zero, the stronger the privacy guarantees.

These results should be interpreted in light of several important limitations. First, the Google mobility data is limited to smartphone users who have opted into Google's location history feature, which is off by default. These data may not be representative of the population as whole, and furthermore their representativeness may vary by location. Importantly, these limited data are only viewed through the lens of differential privacy algorithms, specifically designed to protect user anonymity and obscure fine detail. Moreover, comparisons across rather than within locations are only descriptive since these regions can differ in substantial ways.

**Flight data.** We used data from the International Air Transport Association (https://bluedot.global/) on the monthly number of confirmed passengers on flights (direct and indirect) from India to all other countries from January 2021 to June 2021.

**Estimated importation intensity.** We estimated the weekly importation intensity of the Delta variant for each destination location at the weekly level using the human mobility, GISAID and COG-UK genomic data and COVID-19 case data. An importation intensity value was calculated for each international movement by multiplying the proportion of Delta in the location of origin, the total number of new weekly reported COVID-19 cases and the movement intensity between each origin location and the destination location. We then aggregated all importation intensity values by week and destination location to obtain the EII.

**Estimated exportation intensity.** We estimated the exportation intensity of the Delta variant for each location of origin at the weekly level using aggregated human mobility, genomic and case count data. An exportation intensity value was calculated for each international movement by multiplying the proportion of Delta in the country of origin, the total number of new weekly reported cases and the movement intensity between the country of origin and the destination country. We then aggregated all importation intensity values by week and origin location to obtain the EEI.

**Estimated local human mobility intensity.** To obtain an estimate of the intensity of human mobility within a location, we calculated a 'relative self-mobility' value indicating the intensity of mobility within a location (where the origin and destination of the trips are the same) as a percent of the highest recorded of movement within this location in our mobility data during the time period from 22 March 2020 to 6 June 2021 using the human mobility data described above.

**New Delta lineage introductions.** Daily new lineage introductions into the UK by UTLA were obtained from the continuous phylogenetic analysis described above. The data were aggregated by week and UTLA.

**Statistical modelling of Delta growth**
Data pre-processing: we kept data starting from the 13th (week commencing 28th March 2021) epidemiological week. These dates are referred to as baseline elsewhere in the main text. We excluded weeks after the first time 95% of samples were observed to be Delta in each UTLA because, after this point, we can no longer estimate the relative growth rates reliably since Delta is effectively fixed in the population. Finally, we kept only those UTLAs which had data on Delta for at least 9 weeks (which are not required to be consecutive). In the final dataset, we had 683 observations (across 64 UTLAs with approximately 11 weeks of non-missing data on average for each) (Supplementary Table 8).

**Model.** In what follows, we model the dynamics of Delta penetration within a UTLA. Here, we model how the number of Delta samples per UTLA ($i$), varies over time ($t$) (here measured in weeks). The background transmission conditions driving the observed number of Delta samples in a given UTLA may be similar to other UTLAs within the same region. We model this variation hierarchically and index variables at the UTLA level by $i[j]$ to indicate that UTLA $i$ is nested within (the overarching) NUTS1 unit $j$. We use a binomial sampling distribution to model the number of Delta samples $Z_t^{i[j]}$,

$$Z_t^{i[j]} \sim \text{binomial}(Y_t^{i[j]}, p_t^{i[j]}),$$

where $Y_t^{i[j]}$ is the total number of sequenced samples, and $0 \leq p_t^{i[j]} \leq 1$ is the corresponding proportion of Delta samples in subregion $i$ in week $t$. We then transform this probability, so that it is on the (unconstrained) logit scale:

$$\phi_t^{i[j]} = \text{logit}(p_t^{i[j]}).$$

A key quantity of interest is the relative growth in the proportion of Delta on the logit (that is, log odds) scale, which we estimate weekly and is denoted by $\rho_t^{i[j]}$, where

$$\phi_t^{i[j]} = \phi_{t-1}^{i[j]} + \rho_{t-1}^{i[j]}.$$

Relative growth for each UTLA is modelled spatially as depending hierarchically on its containing region, $j$. It is also assumed to depend on UTLA-specific covariates:

$$\rho_t^{i[j]} = \rho_t^j + \boldsymbol{\beta}' x_t^{i[j]} + \delta_t^{i[j]},$$

where $\rho_t^j$ is a NUTS1-region-level growth trend, $x_t^{i[j]}$ is a vector of covariates, and $\delta_t^{i[j]}$ is a UTLA- and week-specific term representing the deviation from the region-level growth. To account for temporal autocorrelation in the relative growth rate, a given region's relative growth is assumed to follow a random walk centred around its relative growth in the previous week:

$$\rho_t^j \sim \text{normal}(\rho_{t-1}^j, \sigma_2).$$

To assess the importance of covariates, we compared the predictive performance of models which included different sets of covariates. All covariates were standardized by subtracting the mean and dividing by the standard deviation. Since the cumulative proportions vaccinated (considering either the cumulative proportion vaccinated with a 1st or 2nd dose) increased monotonically throughout the time period of observation, we included the UTLA-level mean of these variables

in our regressions: that is, we used time-invariant regressors. We did so to avoid the risk of spurious association due to both Delta and proportions vaccinated growing coincidently. Covariates were chosen as important predictors if including them in the model improved the model fit on a hold-out set comprising the last two weeks of data for each UTLA. Our best model included within-UTLA mobility and time since baseline, which outperformed the model where we included no covariates (Supplementary Table 6). A model including the cumulative proportion vaccinated with a second dose, time since baseline and within-UTLA mobility also outperformed the no covariate model. However, the improvement in prediction accuracy was minimal, and this was the only model outperforming the no covariate model which included vaccinations, so we do not take this as strong evidence of the importance of vaccination in slowing Delta growth.

We estimated our model in a Bayesian framework and chose priors (Supplementary Table 9) so that a wide range of possible Delta proportions were possible yet were centred on low values in the absence of further information: our prior predictive distributions in Extended Data Fig. 11 illustrate these characteristics.

The computations were done using R and Stan using four parallel chains with 50,000 to 60,000 iterations (depending on the model), half of which were discarded as warm-up iterations; the chains were subsequently thinned by a factor of 10. In all cases, MCMC sampling was diagnosed as converged with $\hat{R} < 1.01$, and bulk and tail effective sample sizes >400 for all parameters. For 6 of 15 models used for model comparison (which included neither the no covariate model nor the best fit model), there remained 2 out of 4,410 parameters which had $\hat{R} > 1.01$ or had a tail effective sample size below 400; in all cases, the bulk effective sample sizes exceeded 400. In these models, the last two weeks were held-out from each UTLA to perform out of sample predictions, resulting in a smaller dataset, which likely explains the difficulty in obtaining convergence with 50,000 iterations.

Our model outputted two sets of key quantities: the weekly relative growth rate of Delta over time ($\rho_t^{i[j]}$) and the estimated 'effect' of a variable on Delta growth ($\beta$). To determine the implications of the effect sizes, we computed the estimated proportion of Delta samples when the covariates took factual versus counterfactual values. We considered counterfactual scenarios for within-UTLA mobility, holding all other covariates at their factual values. The counterfactual scenario we considered was:
- Minimum mobility (within-UTLA mobility = 0)
- Maximum mobility (within-UTLA mobility = 1)

The results of these counterfactual simulations are shown in Extended Data Fig. 10.

Simulation and model robustness: to test model parameter identifiability, we performed inference on simulated data. We fixed the parameters and simulated from the model to create hypothetical data (with 5 regions, each with 6 sub-regions (that is, UTLAs) and 15 time points). We then used these data to estimate the known parameters. We were reasonably able to recover our parameters, and the model converged with $\hat{R} < 1.01$, bulk and tail effective sample sizes >400 after 20,000 iterations, discarding 10,000 warm-up iterations and thinning by a factor of 10 (Extended Data Fig. 12 and Supplementary Table 7).

**Reporting summary**
Further information on research design is available in the Nature Research Reporting Summary linked to this article.

## Data availability
UK genome sequences used were generated by the COVID-19 Genomics UK consortium (COG-UK, https://www.cogconsortium.uk/). Data linking COG-IDs to location have been removed to protect privacy, however if you require this data please visit https://www.cogconsortium.uk/contact/ for information on accessing consortium-only data.

The Google COVID-19 Aggregated Mobility Research Dataset used for this study is available with permission from Google LLC. Shapefiles for county-level analyses in the UK are openly accessible via the Global Administrative Database (https://gadm.org/).

## Code availability

Code to reproduce the statistical analyses on Delta growth can be found at https://github.com/sumalibajaj/Delta-Statistical-analysis-share. The code and accession ids of sequences used to run the phylogenetic analysis as well as a GISAID acknowledgment table are available at https://github.com/COG-UK/Delta-analysis.

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

**Acknowledgements** COG-UK is supported by funding from the Medical Research Council (MRC) part of UK Research and Innovation (UKRI), the National Institute of Health Research (NIHR) (grant code: MC_PC_19027), and Genome Research Limited, operating as the Wellcome Sanger Institute. M.U.G.K. acknowledges support from a Branco Weiss Fellowship, Reuben College Oxford, Google.org, the Foreign, Commonwealth and Development Office and Wellcome (225288/Z/22/Z), and The Rockefeller Foundation. S.D. and M.U.G.K. acknowledge support from the European Union Horizon 2020 project MOOD (grant agreement number 874850). O.G.P., M.U.G.K., L.d.P. and A.E.Z. acknowledge support from the Oxford Martin School. V.H. was supported by the Biotechnology and Biological Sciences Research Council (BBSRC) (grant number BB/M010996/1). S.D. is supported by the Fonds National de la Recherche Scientifique (FNRS) (Belgium). J.T.M., R.C. and A.R. acknowledge support from the Wellcome Trust (Collaborators Award 206298/Z/17/Z—ARTIC network). A.R. is also supported by the European Research Council (grant agreement number 725422—ReservoirDOCS) and Bill and Melinda Gates Foundation (OPP1175094—HIV-PANGEA II). C.R. was supported by a Fondation Botnar Research Award (programme grant 6063). G.B. acknowledges support from the Research Foundation—Flanders (Fonds voor Wetenschappelijk Onderzoek—Vlaanderen) (G0E1420N and G098321N) and from the Interne Fondsen KU Leuven (Internal Funds KU Leuven) under grant agreement C14/18/094. A.O. is supported by the Wellcome Trust Hosts, Pathogens and Global Health Programme (grant number 203783/Z16/Z) and Fast Grants (award number 2236). S. Bajaj is supported by the Clarendon Scholarship, University of Oxford and NERC DTP (grant number NE/S007474/1). M.A.S. acknowledges support from US National Institutes of Health grant R01 AI153044. X.J. acknowledges support from US National Institutes of Health grant U19 AI135995. T.P.P. and W.S.B. acknowledge support from the G2P–UK National Virology Consortium funded by the MRC (MR/W005611/1). I.I.B. is supported by the Canadian Institutes of Health Research (grant 02179-000). N.R.F. acknowledges support from the Wellcome Trust and Royal Society Sir Henry Dale Fellowship (204311/Z/16/Z), Bill and Melinda Gates Foundation (INV-034540), the Medical Research Council-Sao Paulo Research Foundation (FAPESP) CADDE partnership award (MR/S0195/1 and FAPESP 18/14389-0) and the MRC Centre for Global Infectious Disease Analysis (reference MR/R015600/1). E.V. acknowledges support from the Wellcome Trust (220885/Z/20/Z). The contents of this publication are the sole responsibility of the authors and do not necessarily reflect the views of the European Commission or any of the other funders.

**Author contributions** J.T.M., V.H., S. Bajaj, O.G.P., A.R. and M.U.G.K. conceived and planned the research. J.T.M., V.H., S. Bajaj, R.E.P., R.I., C.R., C.H., I.I.B., O.G.P., A.R. and M.U.G.K. analysed and processed the data. S. Bajaj performed statistical epidemiological analyses. B.C.L., M.U.G.K., E.V., S. Bhatt, S.D., G.B., X.J. and M.A.S. advised on statistical methodology. J.T.M., V.H., B.J., R.C., A.O., N.J.L., D.M.A. and A.R. designed and implemented genomic data processing pipelines. J.T.M., V.H., O.G.P., S. Bajaj, R.E.P., A.R. and M.U.G.K. wrote the first draft of the manuscript. All authors contributed to editing and interpreting the results. A.R., O.G.P. and M.U.G.K. jointly supervised the work.

**Competing interests** The authors declare no competing interests.

**Additional information**
**Correspondence and requests for materials** should be addressed to Oliver G. Pybus, Andrew Rambaut or Moritz U. G. Kraemer.

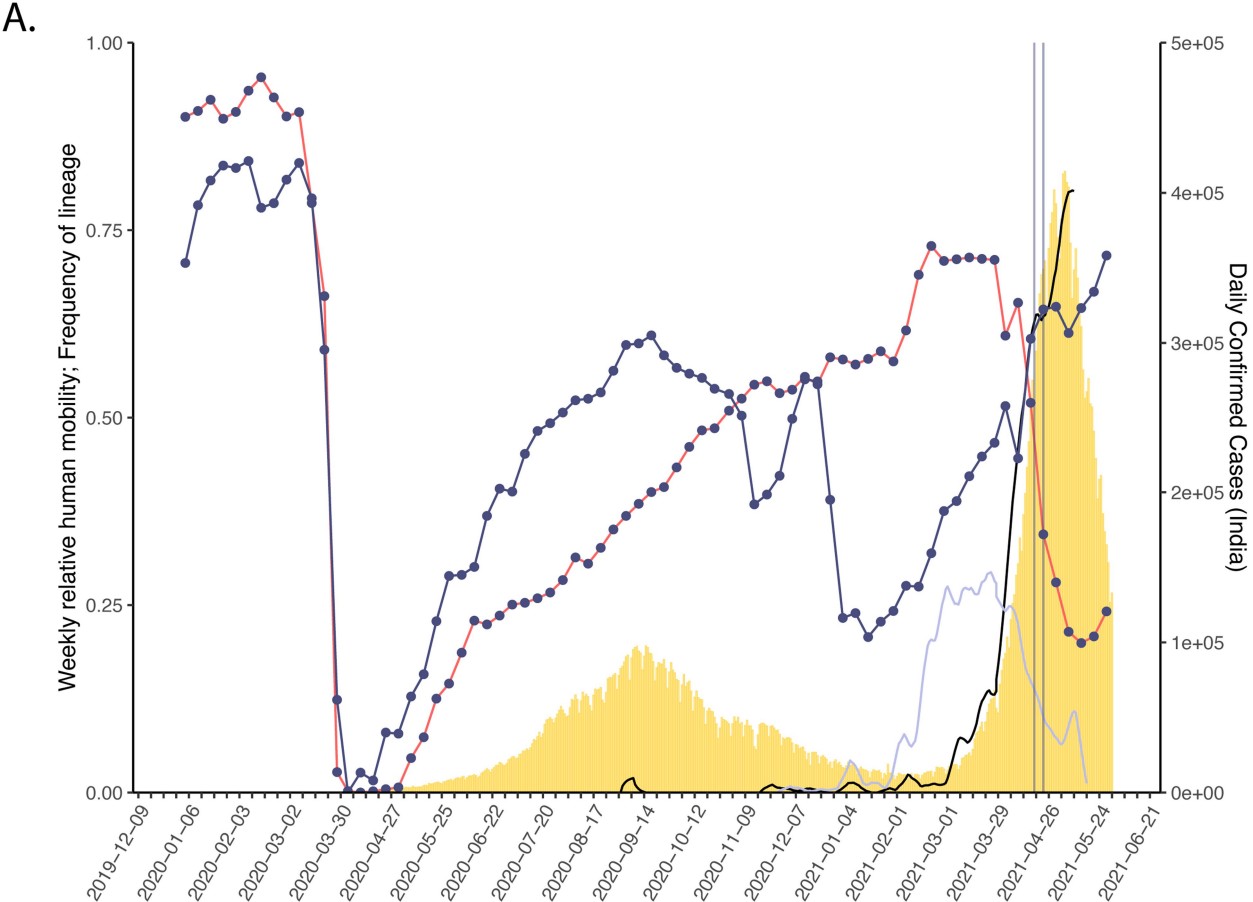

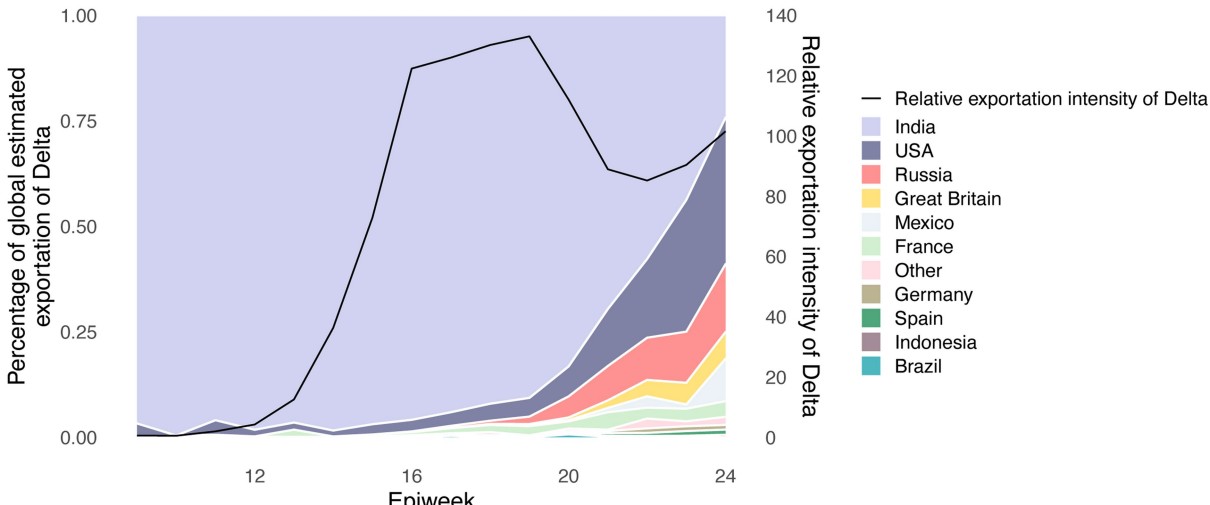

**Extended Data Fig. 1 | SARS-CoV-2 movement dynamics. A**) Daily number of reported SARS-CoV-2 cases (yellow bars, right hand axis) in India. Weekly human movements in England, relative to the maximum in England (dark blue line, left hand axis, Methods), and in India, relative to the maximum in India (red line, left hand axis, Methods). Proportion of genomes in India that are assigned to lineages B.1.617.2 (black line, no points) and B.1.617.1 (light blue line, no points) (left hand axis). First vertical line represents the announcement of the quarantine policy for arrivals of travellers from India to England (17 March 2021) and the second vertical line represents the date of implementation (23 March 2021). **B**) Proportion of weekly Estimated Exportation Intensity (EEI) of Delta by country. See Methods for details of calculation (left y-axis). The black line represents the total EEI by week (right y-axis).

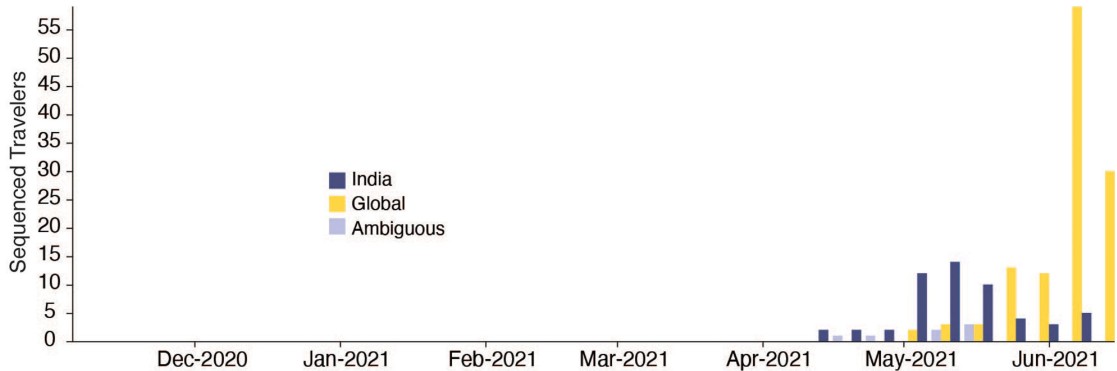

**Extended Data Fig. 2 | Travel history of Delta importations.** Temporal distribution of genomic isolates from the AY.4 sublineage with travel history, by the likely location of exposure.

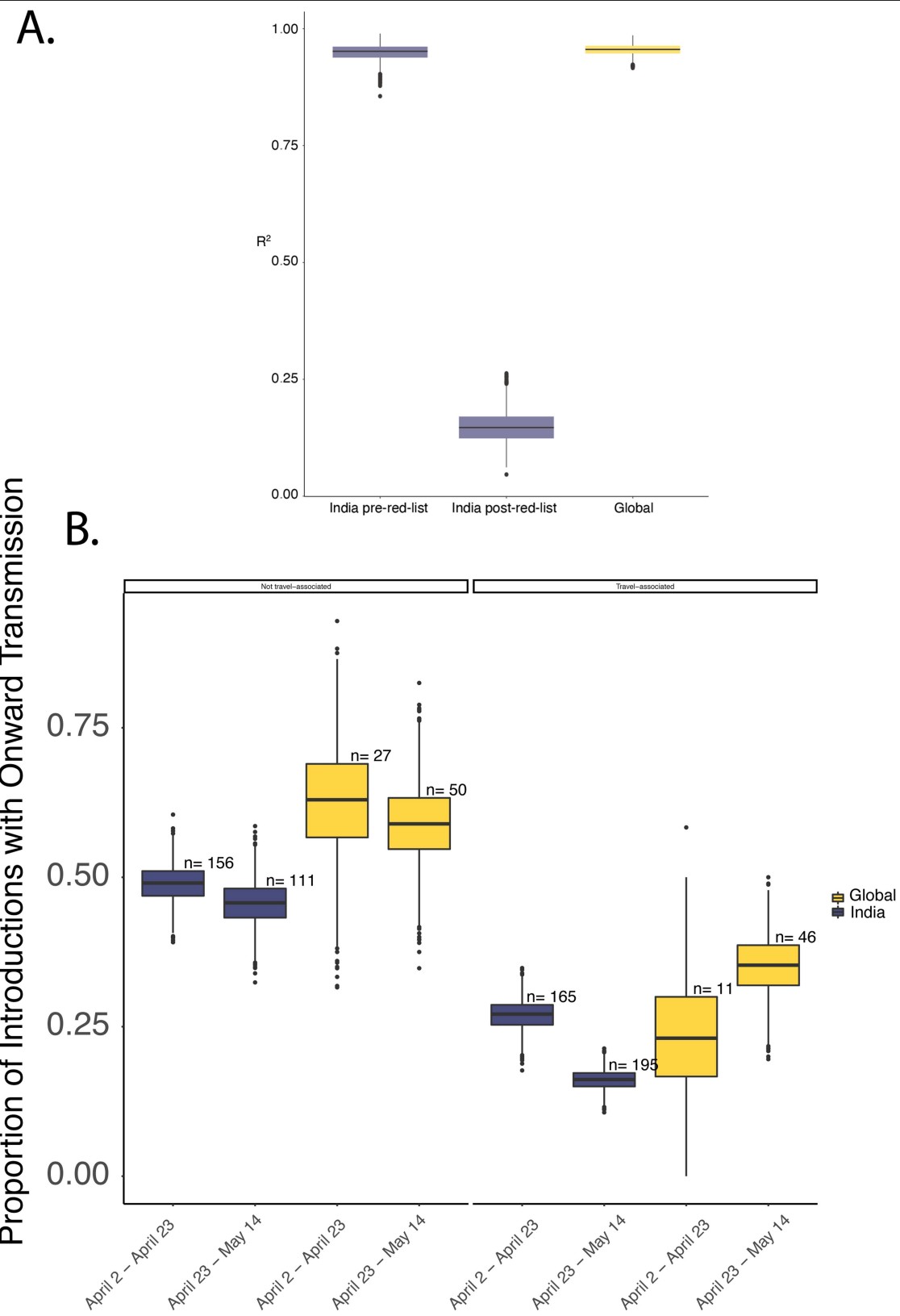

**Extended Data Fig. 3 | Changing importation dynamics. A**) $R^2$ (coefficient of determination) between estimated number of importations from the phylogenetic analysis and the Estimated Importation Intensity (EII) (Fig. 2a). The $R^2$ is calculated separately for India (blue) before and after hotel quarantine was introduced, and for other countries (yellow), whilst also accounting for phylogenetic uncertainty. **B**) Proportion of singletons vs non-singletons stratified by non-travel associated clusters and travel associated clusters and origin locations for the 3 weeks before and after the implementation of the hotel quarantine. The box plot displays the median with lower and upper hinges representing the 25th and 75th percentiles of each group. Whiskers extend to the most extreme data points no more than 1.5 times the interquartile range beyond each hinge. Figure 3a includes each of the trees in the posterior sample (n = 1998 for each box). The number of observations in each group in B is annotated above each box.

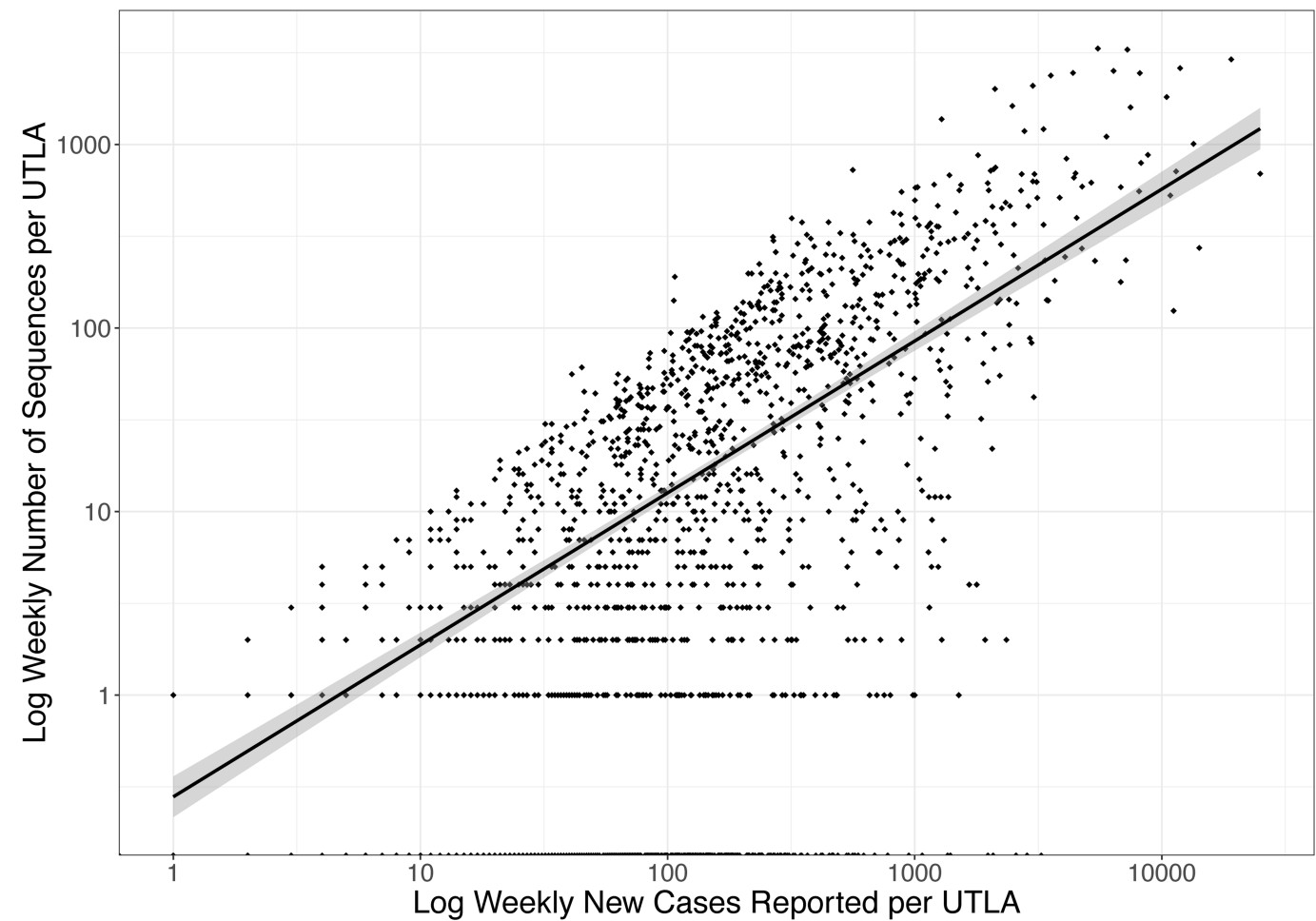

**Extended Data Fig. 4 | Representativeness of SARS-CoV-2 Delta genomes.** Correlation between the number of weekly Delta sequences and the number of weekly Delta cases each UTLA (Pearson's r = 0.68, 95% CI: 0.65–0.71). Shaded region represents 95% CI.

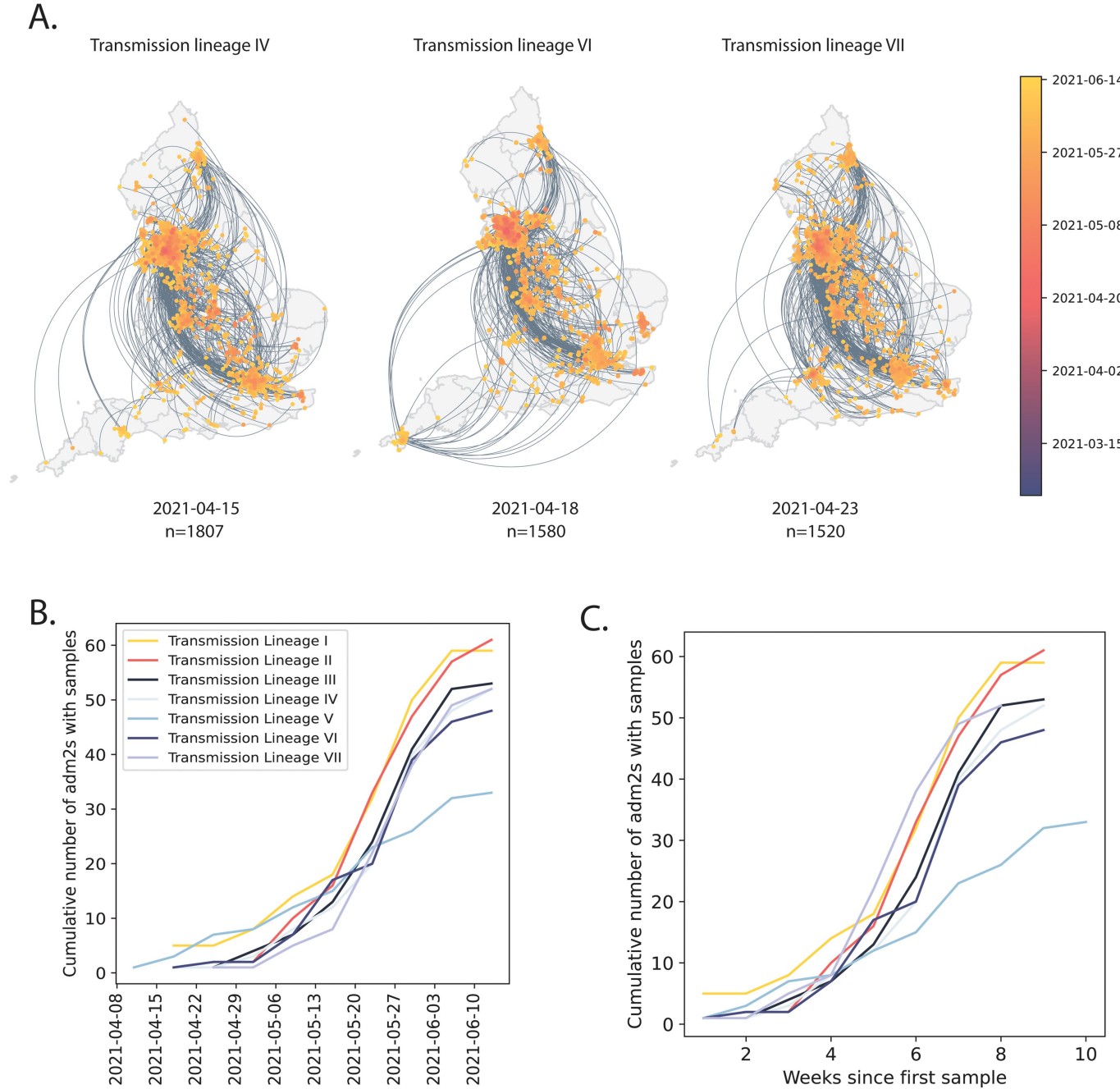

**Extended Data Fig. 5 | Internal dissemination of Delta. A**) Maps showing virus movements inferred using continuous phylogeographic analysis for the fourth, sixth and seventh largest transmission lineages. Direction of movement is anti-clockwise, and dots are coloured by date. Cumulative number of UTLAs that the five largest Delta transmission lineages are sampled in absolute (**B**) and relative (**C**) time.

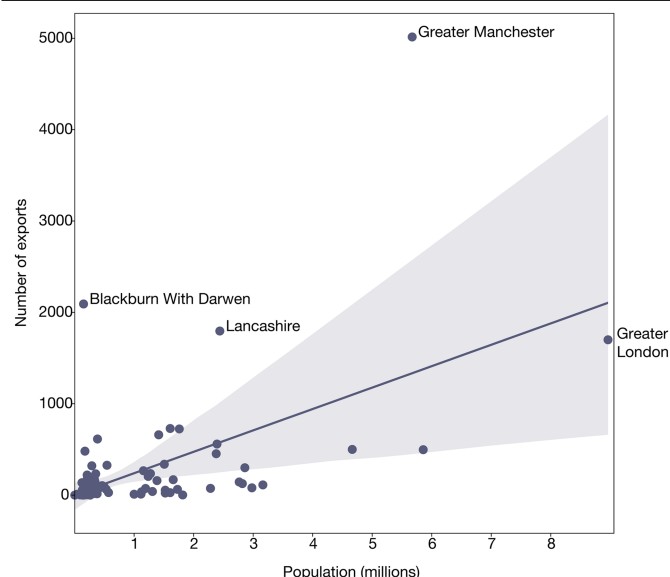

**Extended Data Fig. 6 | Drivers of exportations.** Number of estimated exportations from phylogeographic analysis and population size at the UTLA level (Pearson's r = 0.54, 95% CI: 0.38–0.68, p-value < 0.005). Grey shaded region represents 95% CI.

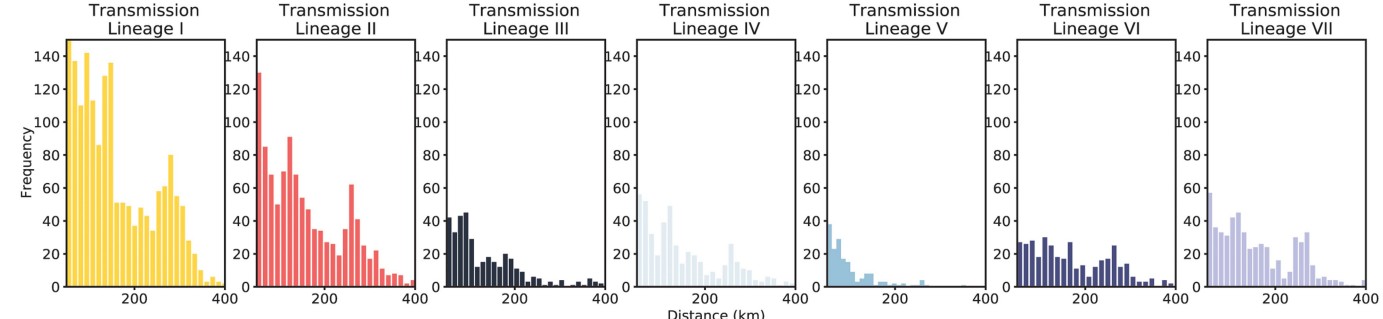

**Extended Data Fig. 7 | Distance of viral movements.** Histograms of the distance of viral movements over 50km for each of the largest seven Delta transmission lineages in England.

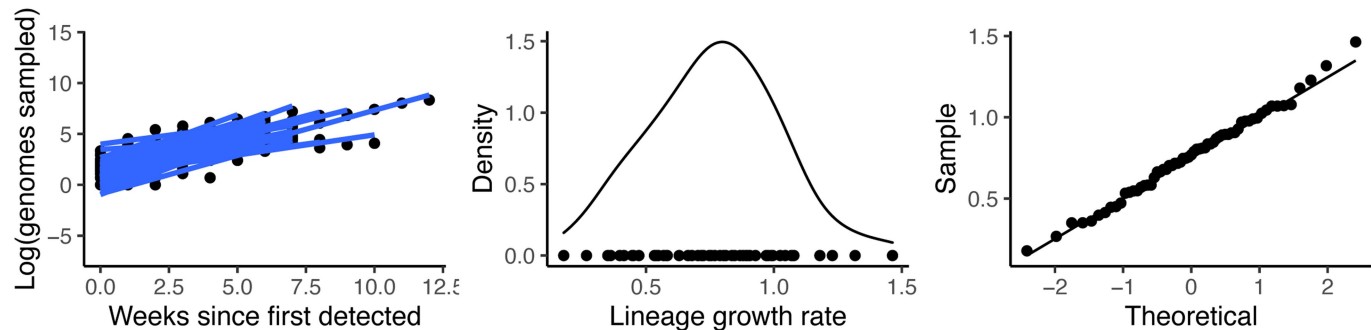

**Extended Data Fig. 8 | Weekly lineage growth rates.** Growth of transmission lineages in England for lineages observed for at least 3 weeks and with >100 genomes sampled in total. **A**) The log number of weekly sampled genomes per transmission lineage plotted over time. Lines represent a linear fit (assuming exponential growth). **B**) Distribution of growth rates (slopes in A). **C**) Quantile plot comparing the observed quantiles in the growth rate distribution to theoretical quantiles from a normal distribution.

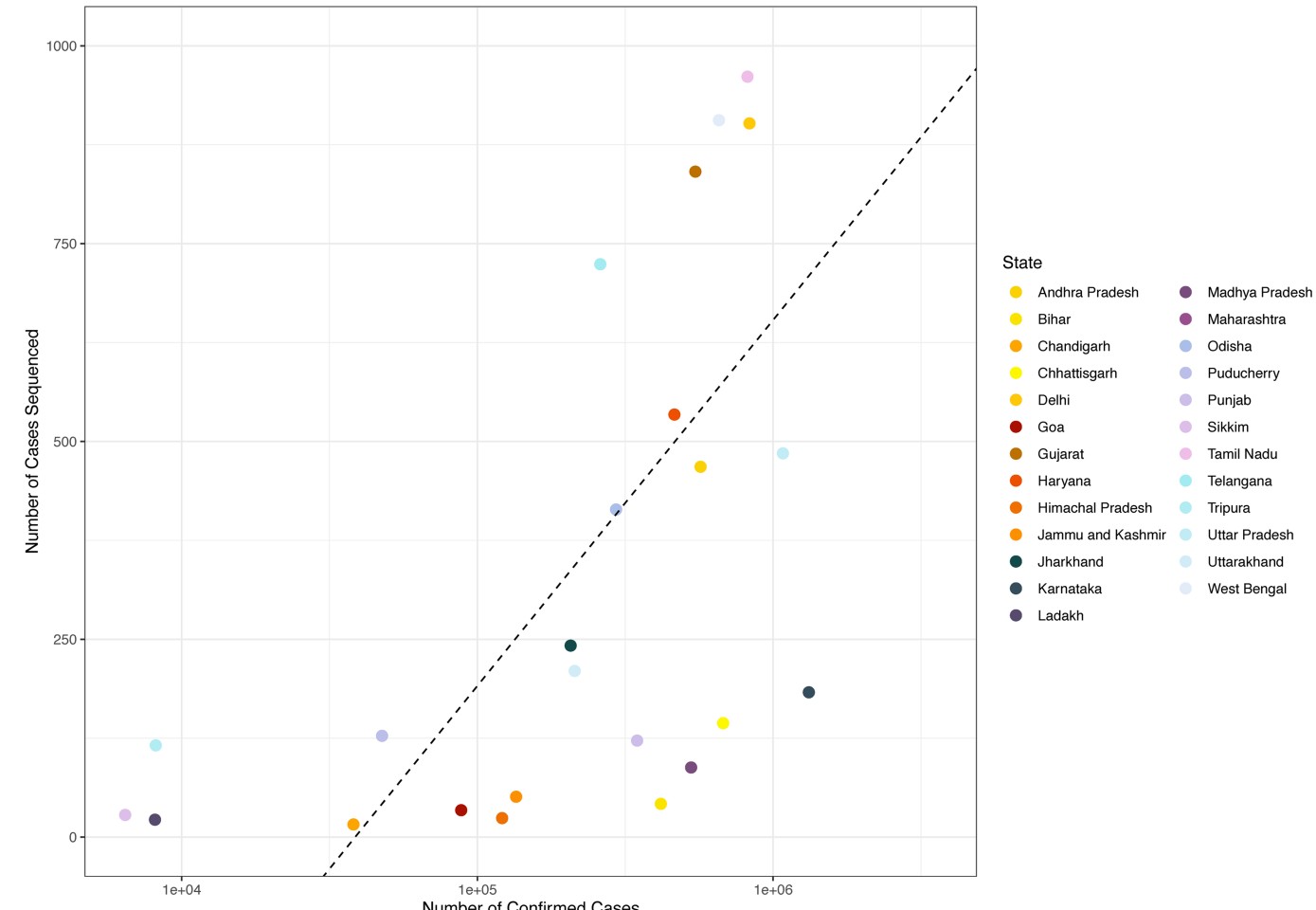

**Extended Data Fig. 9 | Representativeness of SARS-CoV-2 Delta genomes in India.** Scatter plot showing the number of confirmed cases per state in India vs. the number of cases sequenced in that state in India between 28th of November 2020 to the 16th of May 2021. In states above the line more than the mean number of cases were sequenced.

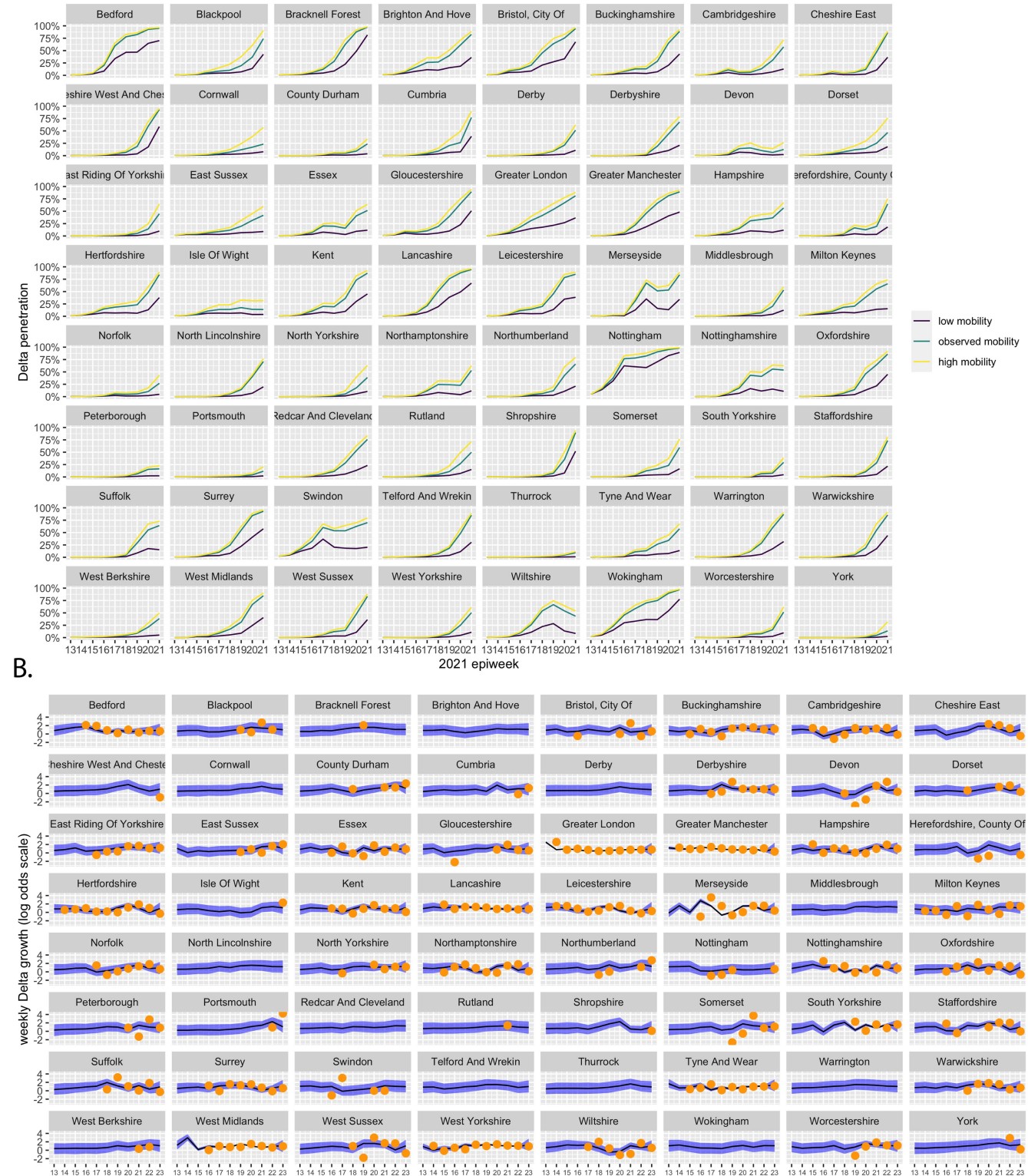

**Extended Data Fig. 10 | Counterfactual scenarios. A)** Estimated and observed proportions of Delta variant samples across UTLAs, for various counterfactual scenarios: minimum (purple) and maximum relative within UTLA mobility (yellow), observed within UTLA mobility (light blue). All lines represent median posterior estimates using the model with epiweek and within UTLA mobility as covariates which was fit to data from 2021 epiweeks 13–21.

**B)** Time-varying relative growth of Delta (on the log odds scale). The shaded regions represent the corresponding 95% Bayesian credible intervals. The orange points indicate the raw growth rates calculated directed from the data (note, there are missing values in these data due to the presence of dates when no samples were taken or when the frequency of Delta remained at 0% or 100%).

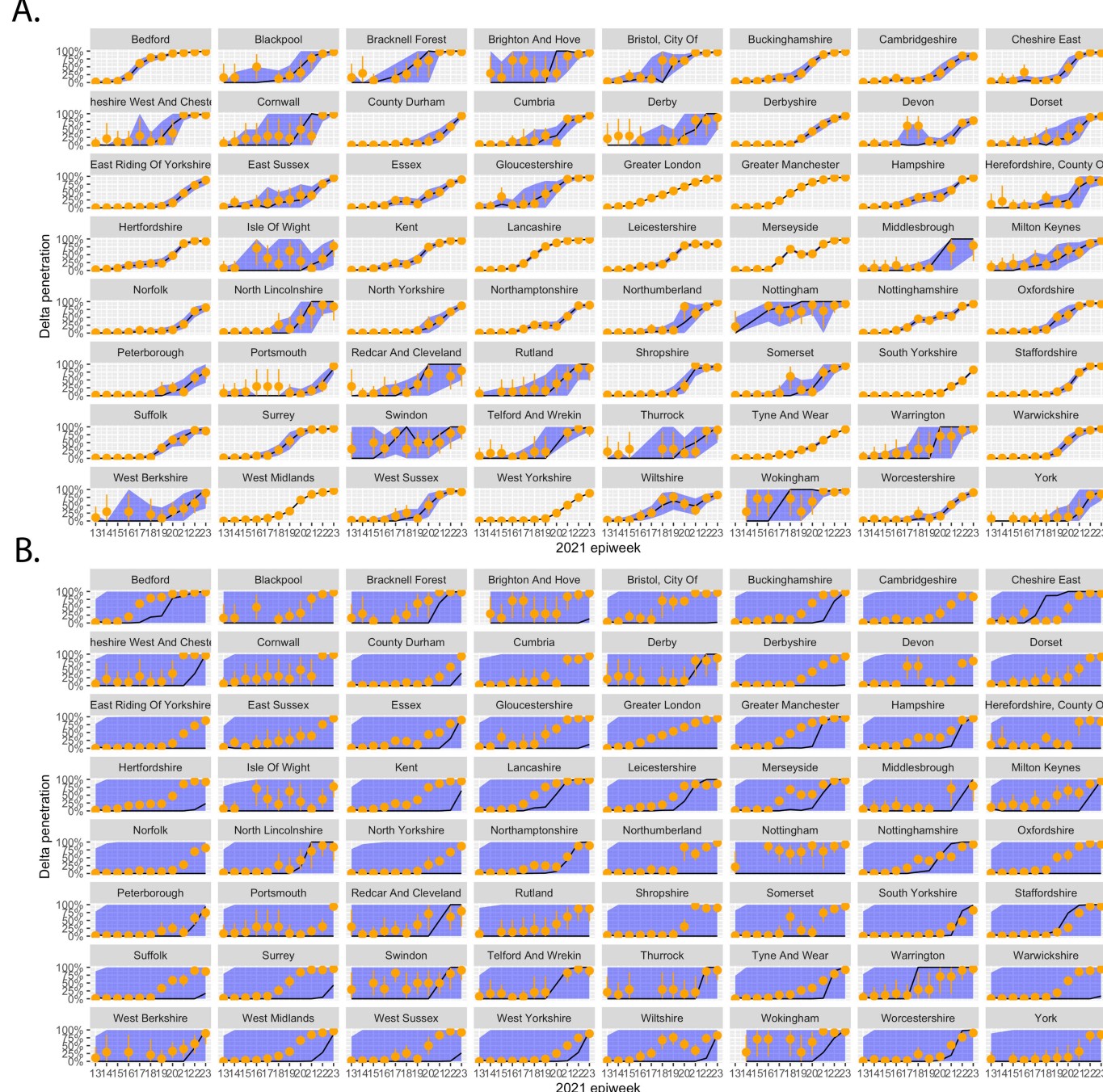

**Extended Data Fig. 11 | Model evaluation. A**) Posterior and **B**) prior predictive simulations for the Delta frequency. In both cases, the orange point-ranges display the observed data (points indicate posterior medians and whiskers indicate 2.5%–97.5% posterior quantiles assuming a uniform prior for each independent data point). The blue shaded region represents in (A) the posterior simulated 2.5%–97.5% quantiles and in (B) the equivalent prior simulated quantiles; lines indicate median simulation values.

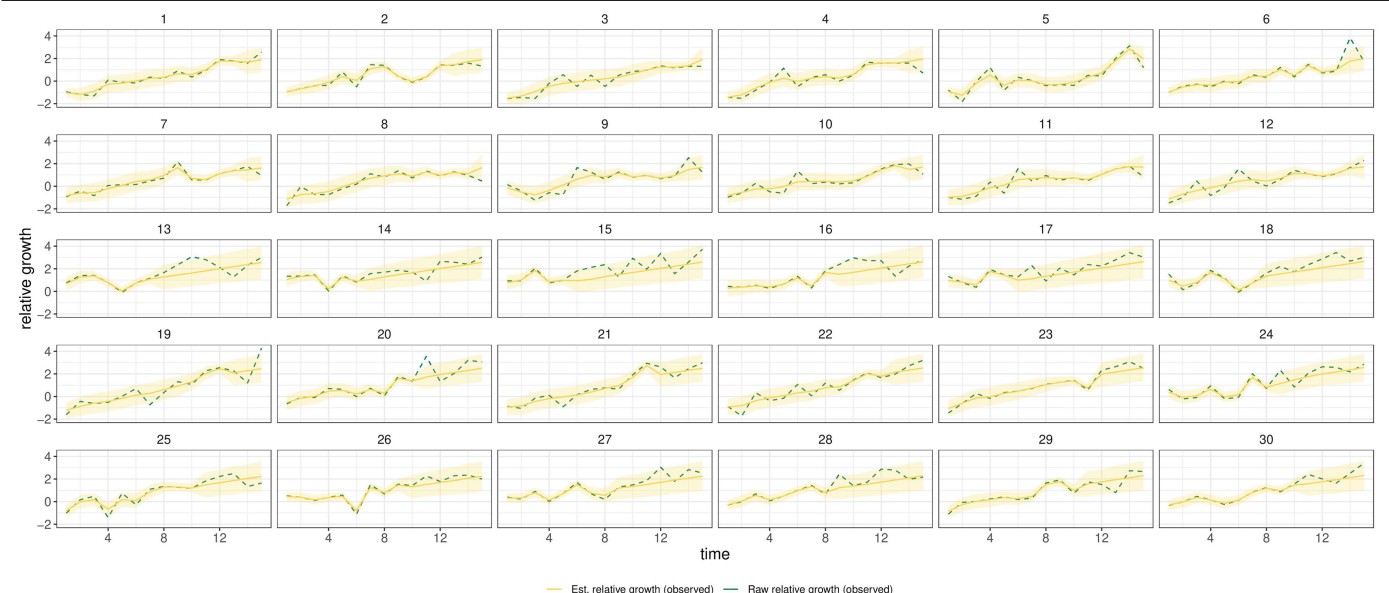

**Extended Data Fig. 12 | Simulation model.** Simulation comparing known vs estimated relative growth rates (see Methods) for hypothetical locations.

# Reporting Summary

## Statistics

For all statistical analyses, confirm that the following items are present in the figure legend, table legend, main text, or Methods section.

| n/a | Confirmed | |
|---|---|---|
| ☐ | ☒ | The exact sample size (*n*) for each experimental group/condition, given as a discrete number and unit of measurement |
| ☒ | ☐ | A statement on whether measurements were taken from distinct samples or whether the same sample was measured repeatedly |
| ☐ | ☒ | The statistical test(s) used AND whether they are one- or two-sided<br>*Only common tests should be described solely by name; describe more complex techniques in the Methods section.* |
| ☐ | ☒ | A description of all covariates tested |
| ☒ | ☐ | A description of any assumptions or corrections, such as tests of normality and adjustment for multiple comparisons |
| ☐ | ☒ | A full description of the statistical parameters including central tendency (e.g. means) or other basic estimates (e.g. regression coefficient) AND variation (e.g. standard deviation) or associated estimates of uncertainty (e.g. confidence intervals) |
| ☐ | ☒ | For null hypothesis testing, the test statistic (e.g. *F*, *t*, *r*) with confidence intervals, effect sizes, degrees of freedom and *P* value noted<br>*Give P values as exact values whenever suitable.* |
| ☐ | ☒ | For Bayesian analysis, information on the choice of priors and Markov chain Monte Carlo settings |
| ☐ | ☒ | For hierarchical and complex designs, identification of the appropriate level for tests and full reporting of outcomes |
| ☐ | ☒ | Estimates of effect sizes (e.g. Cohen's *d*, Pearson's *r*), indicating how they were calculated |

*Our web collection on statistics for biologists contains articles on many of the points above.*

## Software and code

Policy information about availability of computer code

| Data collection | *Provide a description of all commercial, open source and custom code used to collect the data in this study, specifying the version used OR state that no software was used.* |
|---|---|
| Data analysis | Code to partially reproduce the statistical analyses on Delta growth can be found here: https://github.com/sumalibajaj/Delta-Statistical-analysis-share. The code and accession ids of sequences used to run the phylogenetic analysis as well as an GISAID acknowledgment table are available here: https://github.com/COG-UK/Delta-analysis.<br>Explicit details about the analyses and their dependencies can be found in the repositories listed above. Details regarding COGUK genomic processing pipeline can be found at https://github.com/COG-UK/datapipe. In particular these analyses use the following software.<br>minimap2v2.17<br>FastTreev2.1.10<br>BEASTv1.10.4 and BEASTv1.10.4 (commit:d1a45)<br>Tracer v1.7 |

For manuscripts utilizing custom algorithms or software that are central to the research but not yet described in published literature, software must be made available to editors and reviewers. We strongly encourage code deposition in a community repository (e.g. GitHub). See the Nature Portfolio guidelines for submitting code & software for further information.

## Data

Policy information about availability of data

All manuscripts must include a data availability statement. This statement should provide the following information, where applicable:
- Accession codes, unique identifiers, or web links for publicly available datasets
- A description of any restrictions on data availability
- For clinical datasets or third party data, please ensure that the statement adheres to our policy

UK genome sequences used were generated by the COVID-19 Genomics UK consortium (COG-UK, https://www.cogconsortium.uk/). Data linking COG-IDs to location have been removed to protect privacy, however if you require this data please visit https://www.cogconsortium.uk/contact/ for information on accessing consortium-only data. The Google COVID-19 Aggregated Mobility Research Dataset used for this study is available with permission from Google LLC. Code to reproduce the statistical analyses on Delta growth can be found here: https://github.com/sumalibajaj/Delta-Statistical-analysis-share. The code and accession ids of sequences used to run the phylogenetic analysis as well as an GISAID acknowledgment table are available here: https://github.com/COG-UK/Delta-analysis.

# Field-specific reporting

Please select the one below that is the best fit for your research. If you are not sure, read the appropriate sections before making your selection.

☐ Life sciences　　　　☐ Behavioural & social sciences　　　☒ Ecological, evolutionary & environmental sciences

For a reference copy of the document with all sections, see nature.com/documents/nr-reporting-summary-flat.pdf

# Ecological, evolutionary & environmental sciences study design

All studies must disclose on these points even when the disclosure is negative.

| | |
|---|---|
| Study description | Our study characterizes the importation and subsequent spread of the SARS-CoV-2 Delta variant of concern in England. We use the UK genomic data produced by COG-UK to highlight the temporal and spatial dynamics of imported transmission lineages, and novel epidemiological models to determine factors that led to the growth of Delta across both the UK and US. |
| Research sample | The phylogenetic analysis used UK sequences produced by COG-UK and international sequences shared on GISAID. The UK data set consists of pillar 2 English samples taken as part of the national testing infrastructure for community surveillance. These samples were chosen to as they best represent a random sample of lineages circulating in the general English public at the time of study. Only samples identified as the Delta variant of concern were included in this study. UK genome sequences used were generated by the COVID-19 Genomics UK consortium (COG-UK, https://www.cogconsortium.uk/) |
| Sampling strategy | No subsampling strategy was used in this study. All sequences identified as Delta were included. This resulted in a dataset of over 90,000 sequences. This is the largest phylogenetic analysis of it's kind, and used all available pillar 2 English samples generated by COGUK. This was done to generate the best approximation of circulating lineages in England at the time of study.The density of the data allows for detailed modeling of geographic spread in England. |
| Data collection | English samples were taken and sequenced by COG-UK affiliated partners across the UK and shared according to the COG-UK data sharing agreement. Post-processing of genomic data was done as part of the daily COG-UK 'datapipe/phylopipe' hosted on CLIMB. Four sources of data were compiled to provide the travel history for laboratory confirmed cases, depending on availability for each individual case: (1) public health passenger locator forms are required for entry into the UK; (2) routine public health contact tracing data including UK Health Security Agency Second Generation Surveillance System (SGSS)89, (3) COVID-19 test requests with reported travel associations and (4) responses to additional telephone interviews for cases. This data was processed by authors in PHE and shared with others under the appropriate data sharing agreement. We used two human mobility datasets, one a country flight dataset from IATA and another from the Google Mobility Research Dataset which was aggregated to the country level and UTLA level in the UK and state level in the USA. |
| Timing and spatial scale | All SARS-CoV-2 non-UK genomes in GISAID and pillar 2 samples from England that were identified as Delta up until June 15, 2021 were included in this study. The spatial resolution of the global analyses were at the country level and for the detailed phylogeographic analyses at the postcode level. |
| Data exclusions | Sequences with known quality flaws as well as those that failed to pass standard quality measures (e.g. those with low sequence coverage and temporal outliers) were removed from the study. |
| Reproducibility | This study is based on epidemiological data and subsequent modeling. All code to reproduce the results and what data we are able to share has been shared to make reproducing such analysis in this and other settings possible. |
| Randomization | Randomization was not relevant in this study, as it was based on epidemiological observations in the general public and there were no experimental treatments applied. |
| Blinding | Blinding was not relevant to this study design,as it was based on epidemiological observations in the general public and there were no experimental treatments applied. |

Did the study involve field work?　　☐ Yes　　☒ No

# Reporting for specific materials, systems and methods

We require information from authors about some types of materials, experimental systems and methods used in many studies. Here, indicate whether each material, system or method listed is relevant to your study. If you are not sure if a list item applies to your research, read the appropriate section before selecting a response.

## Materials & experimental systems

| n/a | Involved in the study |
|-----|----------------------|
| ☒ ☐ | Antibodies |
| ☒ ☐ | Eukaryotic cell lines |
| ☒ ☐ | Palaeontology and archaeology |
| ☒ ☐ | Animals and other organisms |
| ☒ ☐ | Human research participants |
| ☒ ☐ | Clinical data |
| ☒ ☐ | Dual use research of concern |

## Methods

| n/a | Involved in the study |
|-----|----------------------|
| ☒ ☐ | ChIP-seq |
| ☒ ☐ | Flow cytometry |
| ☒ ☐ | MRI-based neuroimaging |

