## [Peer Review File · Nature]

Manuscript Title: Context-specific emergence and growth of the SARS-CoV-2 Delta variant

Reviewer Comments & Author Rebuttals

Reviewer Reports on the Initial Version:

Referees' comments:

Referee #1 (Remarks to the Author):

This manuscript provides an analysis of genome sequence and other data describing the spread of the Delta variant of SARS-CoV-2, utilising the sequences collected in the UK alongside a larger global dataset. The key results presented are that i) There were a large number of independent introductions of this strain into the UK; ii) The introduction of hotel-based quarantine reduced the risk of introductions into the UK; iii) The Delta variant spread within the UK via travel between regions of the country. iv) The local rate of growth of the Delta variant as a proportion of local cases was associated with levels of local population mixing in regions of the UK, and with levels of population immunity within US states.

Major comments:

1. I did not fully understand the analysis around the claim that introducing hotel-based quarantine reduced the rate of virus importation. This is expressed in terms of the Estimated Importation Intensity (EII), which describes the rate at which infected individuals came into the UK. My expectation for this kind of analysis would be that the authors would look at the rate of inferred introductions divided by the EII, thereby giving a statistic indicating the proportion of imports of the virus into the UK that led to a community introduction. A reduction in this statistic would then indicate that fewer imports led to introductions, consistent with a hypothesis that a given public health measure had reduced this value. However, i) it was not clear that such a statistic was calculated; so far as I could tell, no quantitative value for the extent to which hotel-based quarantine reduced introductions was given. ii) the analysis was instead performed in terms of the correlation between EII and the number of introductions. Here, a correlation between EII and introductions is consistent with people coming into the UK from India and introducing the virus into communities, but a lack of correlation is not uniquely explained by hotel quarantine leading to the reduction of introductions. For example, if the statistics were confounded by measurement errors in one part of the dataset, a lack of correlation might be observed. The link between a loss of correlation the prevention of introductions was not evident to me. Considering the converse situation, might a public health measure that consistently prevented exactly 50% of introductions not lead to an identical correlation between EII and introductions from India? Data following $y=0.5ax+b$ are no less correlated with x than data that follow $y=ax+b$. iii) furthermore, a reliable observation that introductions fell relative to EII after a specific point in time does not imply that the fall was caused by the introduction of one specific public health measure. A variety of other factors, including behavioural change in response to knowledge of a strain spreading due to importation from India is one potential alternative factor here. I may be missing something here.

1.1. A more minor point on introductions. Sequences from the UK are highly represented, with 40% of positive cases sequenced (is it known what proportion of cases were being detected at the time?). By contrast, cases from India are likely to have been dramatically less well sampled. In a measure of introductions where phylogenetic links between sequences denote introductions, what is the likely effect upon this measure of this undersampling? Are the number of introductions under-estimated, or does knowledge of individual travel offset this?

1.1.1. For what proportion of individuals were information about international travel available?

2. A major result in the paper is the finding of a link between local growth rates of Delta and rates of local mixing within the UK. However, i) In Extended Data Table 4 it is clear that the confidence interval for the extent of this link covers zero, so that it is unclear with what level of confidence this conclusion can be drawn. ii) While the result makes some intuitive sense, it is not replicated in the data from the US, with a negative relationship (though with wide confidence intervals spanning zero) between local growth rate and local mixing being observed. Is there a reason why UK and US populations would differ in this regard, or is the result simply a chance occurrence? iii) Other factors have the potential to influence the local growth rate, not least the level of local immunity, for which a positive association is found in the US population. What evidence can be provided that. Could the omission of other potentially unknown factors distort the result presented, if indeed the link found for the UK data can be trusted?

2.1. Although the result here is a major part of the paper, it is not illustrated anywhere in a systematic way. Examples of regions with different local growth rates, or with growth at different time, are shown in Figure 4, but these seem illustrative of a range of differences rather than demonstrative of any underlying causal relationship. For example, a histogram of growth rates might show the range of variation at a given time. The display of results for all of the individual growth curves in supplementary information is commendable, and with study could reveal local differences, but again fails to demonstrate patterns that apply across the set as a whole. Can the results of the regression be shown in a holistic manner?

2.2 In line 292 a comment is made about the beginning of university holidays. I was unclear whether this was a claim of a specific result, or simply a hypothesis about one potential factors underlying difference. It is an interesting claim, but how much effect do university holidays have here, and can these be identified as the specific cause of a higher growth rate in Oxfordshire? Do more people in Oxfordshire go to University than from other regions of the UK?

3. In line 664 of the method section it is noted that some of the posterior set of trees were excluded because they produced a very unlikely outcome, with the root node of subtree 3 in England a long time before the first sample was collected from England. I wonder if this result (i.e. the discovery of unlikely reconstructions in the posterior) arises because there is some information missing to the analysis, leaving the posterior inappropriately unconstrained? If so, would this indicate that the posterior as a whole is distorted by the same lack of information, such that the results as a whole are problematic? Or can these outliers be explained in another manner?

Minor comments:

Figures should be labelled in a consistent way e.g. using markers a) or A both in the figures and in the figure legends.

Line 666: There is a stray word “In” at the end of the sentence.

In the methods section some non-English characters do not display correctly in the PDF. The first instance is on line 624.

Referee #2 (Remarks to the Author):

McCrone et al. has conducted a deep analysis for the early spread of Delta variant from India and its establishment and circulation in UK with high spatio-temporal resolution. They have identified a number of interesting virus transmission dynamics patterns that could help optimize the spatial intervention to control the spread of future VOCs: e.g. focus of geographical expansion of Delta shift from India to global pattern in early May 2021; the hotel quarantine for travellers from India helped to reduce further transmission in UK; increasing inter-regional travel promoted nationwide circulation of Delta; interactions between immunity and human behaviour determine the growth of Delta. They have used huge multi-omic data sets including viral genome sequence, human mobility data and contact tracing data. The methods and models used are excellent and based on strong theory foundation previously published. This is another great work from the group of authors which would further our understanding of COVID-19 transmission and the impact by non-pharmaceutical interventions. I did not find major issues in the study but some minor comments listed below.

Line 71: “Retrospective investigation revealed that Delta was first detected in India in mid-September 2020” - would be useful to provide the source reference for this information. Reference#6,#7 seem did not mention this, and Ref# 5 was suggesting Oct 2020?

Line 99-100: This sentence could be confusing to readers at the first glance, whether 975 is the total number of genomes after subsampling, or it is the number of daily subsamples. I could get the meaning after referring to the other part of the text, but think it can be better re-written.

Line 100-102: Quite complex sentence, two ‘but’. Could be better revised.

Line 103: should be “estimated”, and “most recent common ancestor”.

Line 128-129: Would be useful to add a few words in this paragraph to more explicitly explain what the data provided to achieve the high spatio-temporal resolution analysis.

Line 142, 144, 285, 282, past tense may be better.

Line 151: Would it be easier to understand by directly saying that some countries/regions have

lower sampling insensitivity which could likely lead to underestimate of importations into England?

Line 187: Would be useful to provide a brief explanation there of why it is not probable for unusually long latent period or within-group transmission during the quarantine period, so that readers do not need to jump to the reference#36. In fact, in some places, transmission among residents inside the quarantine hotel was suggested.

Fig 1B, y-axis labels misalignment.

Fig 1C, the y-axis for frequency of delta variant is missing?

Fig 2A, What is the unit of EII? It looks like a weekly estimate but the y-axis indicate a daily frequency. Is it like Fig. 1C where one y-axis is forgotten? Also the yellow line is difficult to see.

Fig 3C and line 236, could the range of viral movements inferred from the phylogeography affected by the variable intensity of sampling in England?

Line 255: 'were not included'

Line 318: 'than some other co-circulating variants' is more accurate.

Line 171-173, 354-356: I feel such comparison and interpretation is quite indirect. Why not directly calculating the ratio between the number of sequences from sporadic introductions by travellers (since the authors have the individual travel data) to the number of local transmission lineage established in UK, and then assess the difference of such ratio before and after the quarantine hotel implementation?

Title: "Context-specific emergence" seems to be very broad - Could it be more specific/precise while keeping the neatness of the title.

In the Discussion section, could the authors provide more in-depth comparison between the findings of Delta here and the patterns/bahaviors of other VOCs emerged/introduced in UK previously published, especially highlighting the impacts by different contexts at different time periods if not contemporary.

Tommy Lam

Referee #3 (Remarks to the Author):

McCrone and colleagues present a large, in-depth investigation into the establishment of the SARS-CoV-2 Delta variant in England in mid-2021. They take advantage of the expansive COG-UK database to capture introductions of Delta into England and track the fate of these introductions. These results have important bearing on non-pharmaceutical interventions as well as how different viral variants fare against preexisting immunity. This latter aspect, however, is the weakest part of the current manuscript.

The authors do an excellent job exploring and explaining why the actual number of Delta introductions into the UK is likely greater than what they have estimated here. I wonder if there is an additional bias at play that has not been explained: repeated introduction of virus identical (or descendent) genomes. In the absence of synapomorphies distinguishing SARS-CoV-2 genomes from UK and India, a Bayesian phylogeographic analysis would tend to group virus from the same country together (without additional information to the contrary). I imagine the migration patterns from the rest of the analysis would inform these probabilities, but my hunch is that, all else being equal, this inference would underestimate the number of introductions when the genomes are uninformative. If this supposition is correct, it would bolster the author's claims that their migration count is indeed biased downwards.

The analysis of pre-existing immunity (whose proxy is estimated using the fraction of people with two vaccine doses) is weak. Further, the comparison to the US is not clearly motivated in the text. In much of the US, one could expect an inverse relationship between vaccine uptake and prior infection, due to regional behavioral difference. Also, public health messaging in the US during the Delta wave did not acknowledge potential for vaccine-breakthrough, so vaccination could have increased risk behavior. Given all the caveats the authors include in this section, they seem to acknowledge these and other weaknesses in this section. Consider removing.

Even if hotel quarantine measures were effective at reducing introductions, it seems the variant was already established in England, rendering moot the impact of restricting additional introduction. Perhaps I misunderstand, but I would appreciate a more explicit interpretation by the authors on the impact of hotel quarantine on the trajectory of the Delta wave in England.

Minor comments:

Page 3, line 97. 'Random' is ambiguous. Do the authors mean to suggest these samples were chosen randomly from all COVID+ samples collected during this time period? Also, since low CT samples are less likely to produce high quality genome sequences, random needs to be qualified.

Page 3, line 99. 'Evenly' should be replaced by 'uniformly', since it presumably refers to a uniform distribution.

Page 4, line 143. Given the sparse sampling in other countries [not India], it seems likely that introductions from low-surveillance countries abroad could be misinterpreted as coming from India. How effective is the travel-aware model (which does an excellent job incorporating travel data) at

accounting for this discrepancy in sampling? It's unclear if the 'global' category (page 21) is capturing both well- and poorly-sampled countries.

Page 7, Regarding the dissemination of Delta in England, it is unclear if urban areas in England were disproportionately sources of dissemination or just areas with large numbers of people.

Page 11, it would be commenting on how similar these dynamics are to the establishment of SARS-CoV-2 at the start of the pandemic (i.e., >1000 introductions). And perhaps worth commenting on how these dynamics are different.

Author Rebuttals to Initial Comments:

Point by point response to reviewers' comments: "Context-specific emergence and growth of the SARS-CoV-2 Delta variant"

Referees' comments:

Referee #1 (Remarks to the Author):

This manuscript provides an analysis of genome sequence and other data describing the spread of the Delta variant of SARS-CoV-2, utilising the sequences collected in the UK alongside a larger global dataset. The key results presented are that i) There were a large number of independent introductions of this strain into the UK; ii) The introduction of hotel-based quarantine reduced the risk of introductions into the UK; iii) The Delta variant spread within the UK via travel between regions of the country. iv) The local rate of growth of the Delta variant as a proportion of local cases was associated with levels of local population mixing in regions of the UK, and with levels of population immunity within US states.

Authors' response: We thank the reviewer for their detailed feedback which we have addressed below.

Major comments:

1. I did not fully understand the analysis around the claim that introducing hotel-based quarantine reduced the rate of virus importation. This is expressed in terms of the Estimated Importation Intensity (EII), which describes the rate at which infected individuals came into the UK. My expectation for this kind of analysis would be that the authors would look at the rate of inferred introductions divided by the EII, thereby giving a statistic indicating the proportion of imports of the virus into the UK that led to a community introduction. A reduction in this statistic would then indicate that fewer imports led to introductions, consistent with a hypothesis that a given public health measure had reduced this value. However, i) it was not clear that such a statistic was calculated; so far as I could tell, no quantitative value for the extent to which hotel-based quarantine reduced introductions was given. ii) the analysis was instead performed in terms of the correlation between EII and the number of introductions. Here, a correlation between EII and introductions is consistent with people coming into the UK from India and introducing the virus into communities, but a lack of correlation is not uniquely explained by hotel quarantine leading to the reduction of introductions. For example, if the statistics were confounded by measurement errors in one part of the dataset, a lack of correlation might be observed. The link between a loss of correlation the prevention of introductions was not evident to me. Considering the converse situation, might a public health measure that consistently prevented exactly 50% of introductions not lead to an identical correlation between EII and introductions from India? Data following $y=0.5ax+b$ are no less correlated with x than data that follow $y=ax+b$. iii) furthermore, a reliable observation that introductions fell relative to EII after a specific point in time does not imply that the fall was caused by the introduction of one specific public health measure. A variety of other factors, including behavioural change in response to knowledge of a strain spreading due to importation from India is one potential alternative factor here. I may be missing something here.

Authors' response: We thank the reviewer for their suggestion. To get a more direct measure of the risk of importation leading to onward transmission in the community we calculate the ratio of inferred

importations that lead to onward transmission before and after the hotel quarantine (onward transmission is here defined as transmission lineages with at least one observed transmission event in England). We find that pre-quarantine 37.7% (95% HPD 34.0%-41.7%) of importations from India led to observed onward transmission. After the implementation of the hotel quarantine only 26.9% (95% HPD 23.0% - 30.2%) of importations from India led to observed onward transmission. In comparison, introductions leading to onward transmission from other locations did not change during the two periods (~50%). We have added a new Figure 2 (see below) and have amended the main text: “We then estimate, from genomic data, the proportion of inferred importations that led to observed onward transmission in the community (defined as at least 1 ancestral node in England) stratified by location of origin in the three weeks pre/post implementation of hotel quarantine. We find that pre-quarantine 37.7% (95% HPD 34.0%-41.7%) of importations from India led to onward transmission. After the implementation, the fraction of importations from India leading to onward transmission dropped (26.9%, 95% HPD 23.0% - 30.2% of importations led to observed onward transmission, Fig. 2c). In comparison, the proportion of introductions from other locations leading to onward transmission did not change during the two periods (~50%, Fig. 2c). The decrease in onward transmission is most apparent in importations associated with travel history, which suggest the trend is driven by the implementation of hotel quarantine and not temporal biases in lineage detection (Extended Data Fig. 5).”

Figure 2: Timing of importations of Delta into England. **A)** Estimated daily number of importations of Delta from India (blue shaded area) and other countries (yellow shaded area), inferred from phylogenetic analysis. Shaded areas show 95% HPDs of the estimate. Blue and yellow lines show the Estimated Importation Intensity (EII) of Delta, obtained by combining data on human movements, cases, and prevalence of Delta, normalized to the same scale as the phylogenetic estimates. Grey vertical lines show the timing of the announcement of travel restrictions from India to England (April 18, 2021) and their implementation on April 23, 2021. **B)** Temporal distribution of genome sequences from cases with known travel history from India (blue) and other countries (yellow). Isolates with recent travel to both India and other countries are considered ambiguous (light purple) **C)** Proportion of all virus introductions that show evidence of onward transmission in the UK, estimated separately for weeks before and after the

implementation of hotel quarantine (April 23, 2021) and stratified by location of origin (India = light purple, other countries = yellow).

1.1. A more minor point on introductions. Sequences from the UK are highly represented, with 40% of positive cases sequenced (is it known what proportion of cases were being detected at the time?). By contrast, cases from India are likely to have been dramatically less well sampled. In a measure of introductions where phylogenetic links between sequences denote introductions, what is the likely effect upon this measure of this undersampling? Are the number of introductions under-estimated, or does knowledge of individual travel offset this?

Authors' response: The reviewer is correct. The number of introductions from a poorly-sampled to a highly-sampled location will be underestimated (during the study period about ¼ of all infections were detected in England¹), since multiple transmission lineages (independent introductions) in the highly-sampled location are likely to be aggregated together (as mentioned on page 5, 2nd paragraph). This means that the 1,458 introductions estimated from our phylogenetic analysis is a lower bound on the true number of introductions (as mentioned on page 5, first paragraph). We may also expect this aggregation to shift the introduction times in the phylogenetic tree to be before the true introduction times. However, we still observe a good correlation between individual travel histories and the inferred introduction times, meaning this shift is likely small or only affecting a small number of lineages. The travel history-aware model does to some extent offset these issues. We provide an in depth discussion of the potential for underestimating importations here².

1.1.1. For what proportion of individuals were information about international travel available?

Authors' response: We have international travel history for 1.4% (770) of the 52,992 English samples. For context, contact tracing in England was performed for all arriving international travellers. During the period where Delta importations occurred, India was on the Amber travel list (PCR testing on days 2 and 8 required for all arriving travellers). This value has been added to the main text on page 5, first paragraph. We have added this information to the main text: “These inferred importation dynamics match closely data on individual travel histories obtained from infected incoming international passengers (origin-destination travel histories are available for 1.4% of genomes; n = 770; Fig. 2b).”

2. A major result in the paper is the finding of a link between local growth rates of Delta and rates of local mixing within the UK. However, i) In Extended Data Table 4 it is clear that the confidence interval for the extent of this link covers zero, so that it is unclear with what level of confidence this conclusion can be drawn. ii) While the result makes some intuitive sense, it is not replicated in the data from the US, with a negative relationship (though with wide confidence intervals spanning zero) between local growth rate and local mixing being observed. Is there a reason why UK and US populations would differ in this regard, or is the result simply a chance occurrence? iii) Other factors have the potential to influence the local growth rate, not least the level of local immunity, for which a positive association is found in the US population. What evidence can be provided that. Could the omission of other potentially unknown factors distort the result presented, if indeed the link found for the UK data can be trusted?

Authors' response: We thank the reviewer for their comments on the statistical model. As per suggestion of the editor and Rev. 3 we have removed the results for the US states. It is not possible to compare the two models given the variation in spatial resolution, public health messaging (comment from Rev. 3), and differences in population behaviour and genomic data collection.

In recent years, there has been a push to evaluate predictors based on their boost to predictive accuracy opposed to their statistical significance, since the former is more likely to be of practical importance (see, for example, Lo et al., 2015)³. Further, we use the Bayesian approach to uncertainty quantification, so we did not perform null hypothesis tests of significance based on variables' credible intervals. Instead, we evaluated predictors based on whether they could improve model accuracy. To do so, we performed cross-validation, where we tested the model on the final few weeks of data, which was not used to train the model.

We have performed additional experiments to examine the robustness of our results for England (see updated Methods section). We systematically evaluate the effect of covariates in simple (one variable) and multivariate models. Potential confounders included were the number of cumulative vaccinations (two separate covariates: cumulative % receiving a 1st dose and cumulative % receiving a 2nd dose) and the number of introductions (in three variations: contemporaneous number introduced; number introduced lagged by one week and number introduced lagged by two weeks). The results of this study are provided in the updated Extended Data Table 6.

In summary, we find that within-UTLA mobility correlates with the speed at which the Delta proportion increases (i.e., lower mobility leads to slower invasion). This finding is intuitive and highlights the continued importance of NPIs in reducing the speed at which new variants can spread. A multivariate model which included within-UTLA mobility, the cumulative % receiving a 2nd vaccine dose and the time since baseline did marginally outperform the no-covariate model, but since no other model which bettered the no-covariate model included this vaccine variable, we do not take this as strong evidence of its importance in shaping Delta growth. Single-variable models that consider the number of importations or cumulative vaccinations did not correlate with the speed of Delta invasion, and we conclude that the local characteristics of mobility in the UTLAs were more important in driving the invasion of Delta.

We have updated the main text to reflect these changes:

“Our model estimates that the most important tested predictor of the variation in growth of Delta (relative to Alpha) across UTLAs in England was within-UTLA mixing (i.e., relative changes in weekly within-UTLA human mobility, compared to the pre-pandemic period, Figure 4a,b, Extended Data Table 5). The importance of within-UTLA mobility as a factor during the emergence of a new variant (until Delta prevalence reaches 85% (25%-75% quantiles: 78%-96%)) is unsurprising, as preemptive restrictions on movement and social mixing slow the emergence of new pathogens or variants⁵ (see counterfactual scenarios in Extended Data Fig. 9); the cost/benefit ratio of such restrictions will of course depend on the specific context of variant emergence. The relaxation of NPIs therefore increased both within- and among-region transmission (see Fig. 3c). Other European countries did not observe such a rapid increase in Delta relative frequency during May 2021⁶; possible reasons for this difference are (i) during that time levels of mobility and mixing (both local and regional) were lower in those countries and/or (ii) those

countries potentially received fewer international importations of Delta (86,489 passengers flew from India to the UK between March and June, whilst 43,515 flew to Germany, and 16,688 to France, during the same time).

To evaluate the importance of the predictors we use model comparison and out-of-sample prediction (withholding data from the final two weeks)⁴. We find that the model that included within-UTLA mobility fits the observed trajectory of Delta relative growth better than all other tested models (Supplementary Information, Extended Data Figs. 13, 14, 15, Extended Data Tables 4, 5). Model fit did not improve when including weekly numbers of independent viral introductions estimated from genomic data or vaccination rates (Extended Data Tables 5).”

We also agree with the reviewer that our model can only examine the association between covariates and Delta growth and that the model is not designed to examine causal relationships. We have added a sentence in the discussion that the results should not be interpreted as causal (see also the response to the reviewer’s comment 2.1). However we do believe that our model extensions are valuable for future work in predicting growth rates of new VOCs, which will increasingly be mediated by the interactions between immunity and behaviour at local scales (for example, Omicron BA.1 has grown rapidly in urban areas whereas BA.2 in the UK seems to impact rural areas⁷).

2.1. Although the result here is a major part of the paper, it is not illustrated anywhere in a systematic way. Examples of regions with different local growth rates, or with growth at different time, are shown in Figure 4, but these seem illustrative of a range of differences rather than demonstrative of any underlying causal relationship. For example, a histogram of growth rates might show the range of variation at a given time. The display of results for all of the individual growth curves in supplementary information is commendable, and with study could reveal local differences, but again fails to demonstrate patterns that apply across the set as a whole. Can the results of the regression be shown in a holistic manner?

Authors’ response: We thank the reviewer and agree that the presentation of the results could be improved. We have added two visualisations (see Figure below) and associated text in the main manuscript:

- a) To illustrate the general trend in differences in relative growth of Delta and its relationship with within-UTLA mobility we first plot the proportion of Delta stratified by within-UTLA mobility (see Figure 4 below). In this plot, we consider the top and bottom 5 UTLAs (with an overall count of sequenced samples equal to or above 500) according to the average level of within-UTLA mobility. The plot shows that the UTLAs with the highest within-UTLA mobility experienced higher growth than those with lower and medium mobility across the study period.
- b) Further, to show the region-specific relationship between relative growth rates of Delta (model output) and within-UTLA mobility we have added a scatterplot in the Supplementary Information (see Extended Data Figure below). Each panel of this plot shows data for a single region and plots the within-UTLA mobility versus weekly change in Delta proportion. Across all regions, there was a positive association between the variables.

Figure 4: Variation in Delta growth rates across UTLAs in England. A) Shows the increase in Delta frequency compared to Alpha at the UTLA level. UTLAs are coloured according to the level of average within-UTLA mobility by (high = 5 UTLAs with the highest within-location mobility, low = 5 UTLAs with the lowest within-location mobility, medium = remaining UTLAs). The solid lines show data for given UTLAs; the dashed lines show loess curves fit to the data for each mobility category. B) Examples of weekly growth rates of UTLAs with high (left) and low (right) within-UTLA mobility. In panel A, we plot only those UTLAs where the number of sequenced samples was 500 or greater.

Extended Data Figure: Regional scatterplots of within-UTLA mobility vs. Delta growth rates. The blue lines indicate (unweighted) linear regressions for each region.

2.2 In line 292 a comment is made about the beginning of university holidays. I was unclear whether this was a claim of a specific result, or simply a hypothesis about one potential factors underlying difference. It is an interesting claim, but how much effect do university holidays have here, and can these be identified as the specific cause of a higher growth rate in Oxfordshire? Do more people in Oxfordshire go to University than from other regions of the UK?

Authors' response: We agree with the reviewer. It was unclear from the presentation of our results. We did not specifically investigate the patterns of University holidays in our model and have therefore removed the sentence from the main results of the manuscript.

3. In line 664 of the method section it is noted that some of the posterior set of trees were excluded because they produced a very unlikely outcome, with the root node of subtree 3 in England a long time before the first sample was collected from England. I wonder if this result (i.e. the discovery of unlikely reconstructions in the posterior) arises because there is some information missing to the analysis, leaving the posterior inappropriately unconstrained? If so, would this indicate that the posterior as a whole is distorted by the same lack of information, such that the results as a whole are problematic? Or can these outliers be explained in another manner?

Authors' response: The text referred to in the methods incorrectly refers to a previous version of the analysis that was not used in the paper. This part of the text in the Methods section has been removed.

Such trees were not removed from the analysis. We apologize for the oversight and have corrected the text.

In general, we agree with the reviewer that the presence of such unlikely reconstructions results from biases inherent in current phylogeographic methods. Our current approach does not account for differences in sampling proportions across locations and is known to incorrectly aggregate recent introductions into fewer, older, larger introductions. For this reason our estimate represents a lower bound of the true number of introductions (see response above). However, despite these biases we find across all realizations our results are consistent with expectations from independent estimates of importation intensity.

Minor comments:

Figures should be labelled in a consistent way e.g. using markers a) or A both in the figures and in the figure legends.

Authors' response: We thank the reviewer and have changed all labels in Figures to A, B, C ... etc.

Line 666: There is a stray word "In" at the end of the sentence.

Authors' response: We have now fixed this typo.

In the methods section some non-English characters do not display correctly in the PDF. The first instance is on line 624.

Authors' response: We have now fixed this issue.

Referee #2 (Remarks to the Author):

McCrone et al. has conducted a deep analysis for the early spread of Delta variant from India and its establishment and circulation in UK with high spatio-temporal resolution. They have identified a number of interesting virus transmission dynamics patterns that could help optimize the spatial intervention to control the spread of future VOCs: e.g. focus of geographical expansion of Delta shift from India to global pattern in early May 2021; the hotel quarantine for travellers from India helped to reduce further transmission in UK; increasing inter-regional travel promoted nationwide circulation of Delta; interactions between immunity and human behaviour determine the growth of Delta. They have used huge multi-omic data sets including viral genome sequence, human mobility data and contact tracing data. The methods and models used are excellent and based on strong theory foundation previously published. This is another great work from the group of authors which would further our understanding of COVID-19 transmission and the impact by non-pharmaceutical interventions. I did not find major issues in the study but some minor comments listed below.

Authors' response: We thank the reviewer for their positive feedback. We have addressed the specific comments below.

Line 71: “Retrospective investigation revealed that Delta was first detected in India in mid-September 2020” - would be useful to provide the source reference for this information. Reference#6,#7 seem did not mention this, and Ref# 5 was suggesting Oct 2020?

Authors’ response: We agree and apologise. There were a few sequences uploaded to GISAID that were backdated to September 2020. However, these were of low quality and did not pass our quality control (which is detailed in the SI). We have removed the sentence from the introduction.

Line 99-100: This sentence could be confusing to readers at the first glance, whether 975 is the total number of genomes after subsampling, or it is the number of daily subsamples. I could get the meaning after referring to the other part of the text, but think it can be better re-written.

Authors’ response: We agree with the reviewer and have edited the sentence which now reads: “To provide global context for the emergence of Delta in the UK, we first conducted a phylodynamic analysis by uniformly subsampling Delta SARS-CoV-2 genome sequences by collection date between March 4, 2021 and June 15, 2021 (n = 975).”

Line 100-102: Quite complex sentence, two ‘but’. Could be better revised.

Authors’ response: We agree with the reviewer and have simplified the sentence: “Details of the origin and spread of Delta within India are still uncertain. A substantial increase in genomic surveillance across the country will likely facilitate the study of the emergence and expansion there but is outside the scope of this work.”

Line 103: should be “estimated”, and “most recent common ancestor”.

Authors’ response: We have made these changes.

Line 128-129: Would be useful to add a few words in this paragraph to more explicitly explain what the data provided to achieve the high spatio-temporal resolution analysis.

Authors’ response: We agree with the reviewer and have now simplified the sentence to read: “Virus genomes were generated from ~40-60% of all positive cases in England during the emergence of Delta between March and May 2021 (Fig. 1c)³⁷ and combined with metadata on the locations (at the Upper Tier Local Authority (UTLA) level), enabling us to trace the virus’ introduction and characterize its spread at a high spatio-temporal resolution³⁷.”

Line 142, 144, 285, 282, past tense may be better.

Authors’ response: We agree with the reviewer and have made these changes.

Line 151: Would it be easier to understand by directly saying that some countries/regions have lower sampling insensitivity which could likely lead to underestimate of importations into England?

Authors' response: We agree and have adapted the sentence which now reads: “High variation in sampling intensity among countries (specifically high sampling intensity in England compared to many other countries) means the true number of importations into England is likely much larger than that inferred from phylogeographic analysis alone (Fig. 1b & c, see related discussion in the context of the UK's first wave²).”

Line 187: Would be useful to provide a brief explanation there of why it is not probable for unusually long latent period or within-group transmission during the quarantine period, so that readers do not need to jump to the reference#36. In fact, in some places, transmission among residents inside the quarantine hotel was suggested.

Authors' response: Since the submission of our work there have been additional reports of onward transmission in hotel quarantine outside of the UK. We have removed the sentence saying that we do not consider it probable: “(ii) individuals may have become infectious and transmitted only after leaving quarantine, either due to an unusually long latent period or within-group transmission during the quarantine period, ~~although we do not consider this probable~~⁹”.

Fig 1B, y-axis labels misalignment.

Authors' response: We thank the reviewer for their observation and have adjusted the axis labels accordingly.

Fig 1C, the y-axis for frequency of delta variant is missing?

Authors' response: We apologize for the confusion. The frequency of the Delta variant is plotted on the same y-axis [0,1] as the proportion of sequenced cases. We have adjusted the figure legend to communicate the relationship more clearly.

Fig 2A, What is the unit of EII? It looks like a weekly estimate but the y-axis indicate a daily frequency. Is it like Fig. 1C where one y-axis is forgotten? Also the yellow line is difficult to see.

Authors' response: EII represents a relative importation intensity that is the product of the estimated number of Delta cases in a location and the relative movement between that location and the UK in arbitrary units. As such, the absolute values are on an arbitrary scale (they could in theory be rescaled between 0 and 1); however, the relative changes in EII through time correctly represent changes in importation rate and when imports are expected to occur, and EII is well correlated with phylogenetic estimates of importation dynamics. We have added a second y-axis to make this more clear.

Fig 3C and line 236, could the range of viral movements inferred from the phylogeography affected by the variable intensity of sampling in England?

Authors' response: We thank the reviewer for their suggestion and have added a plot to the Supplementary materials showing the correlation between weekly SARS-CoV-2 Delta sequences vs.

weekly Delta cases (see below) and have added a reference to the Figure in the main manuscript (see Figure below). Given the high sampling proportion during the time frame of the study (~40% of COVID+ cases) and strong correlation between sequences and reported cases we do not anticipate there to be biases in the phylogeographic inference. We have added text to the main manuscript: Sequence sampling was highly representative of reported cases at the UTLA level (Extended Data Fig. 6), making possible the reconstruction of virus movements across England using continuous phylogeography approaches⁴⁵.

Figure: Correlation between the number of weekly Delta sequences and the number of weekly Delta cases each UTLA (Pearson's $r = 0.68$, 95% CI: 0.65 - 0.71).

Line 255: *'were not included'*

Authors' response: We have now fixed this typo.

Line 318: *'than some other co-circulating variants' is more accurate.*

Authors' response: We agree and have made this change.

Line 171-173, 354-356: *I feel such comparison and interpretation is quite indirect. Why not directly calculating the ratio between the number of sequences from sporadic introductions by travellers (since the authors have the individual travel data) to the number of local transmission lineage established in UK, and then assess the difference of such ratio before and after the quarantine hotel implementation?*

Authors' response: We agree with the reviewer and have calculated a more direct measure of the number of introductions leading to onward transmission in England (defined as transmission lineages of size >1). We find that the number of importations leading to onward transmission from India decreases from 43.7% to 24.5% percent (please also see detailed response to Rev. 1). We have added a new Figure 2C and

edited the main text: “We then estimate, from genomic data, the proportion of inferred importations that led to observed onward transmission in the community (defined as at least 1 ancestral node in England) stratified by location of origin and pre/post implementation of hotel quarantine. We find that pre-quarantine 56.3% (95% HPD 53.0%-59.7%) of importations from India led to onward transmission. After the implementation the fraction of importations from India leading to onward transmission dropped (25.5%, 95% HPD 21.4% -27.6% of importations led to observed onward transmission, Fig. 2c). In comparison, introductions from other locations leading to onward transmission did not change during the two periods (~50%, Fig. 2c). This pattern remained when transmission lineages were stratified by whether they had reported travel history(Extended Data Fig. 5).”

Title: "Context-specific emergence" seems to be very broad - Could it be more specific/precise while keeping the neatness of the title.

Authors' response: We thank the reviewer for their suggestion. After having attempted to expand the title we did not find a suitable solution and believe the title captures the presented work as well as possible given the title length constraints.

In the Discussion section, could the authors provide more in-depth comparison between the findings of Delta here and the patterns/bahaviors of other VOCs emerged/introduced in UK previously published, especially highlighting the impacts by different contexts at different time periods if not contemporary.

Authors' response: We agree with the reviewer and have added a paragraph to the discussion describing the differences in spatio-temporal spread of VOCs in the UK including details on the expansion of Omicron:“In comparison to the Alpha variant, which arose and spread from a single location in South East England¹⁰, the expansion of the Delta variant was predominantly due to exports from the North West (Figure 3). Both the analysis of the Alpha variant and of the first wave of SARS-CoV-2 in spring 2020 suggested that Greater London played an substantial role in spreading SARS-CoV-2 across England, as expected, given it is England's largest city by far , and highly connected by road, rail and air to other locations. However, Greater London was less important in the spread of Delta, even after Delta had become established there. This indicates the importance of founder effects; where a VOC first becomes established within a country may have a strong impact on subnational spatial dissemination, information that is useful for planning localised interventions.

Further, while there are intrinsic differences in transmissibility between VOCs, the role of NPIs and levels of immunity from prior infection or vaccination also impact their dynamics. After the start of the first UK national lockdown during the first wave of infections in 2020, lineage movements were severely curtailed and most lineages went extinct (Du Plessis et al. 2021); in comparison, viral movement of Delta lineages increased after the relaxation of NPIs, accompanied with a subsequent rise in positive cases (Figure 3). For the latest VOC, Omicron, NPIs have remained relatively stable throughout England, and the increase in cases of Omicron BA.2 in more rural areas in the South West in February and March 2022 has been speculated to be due to lower infection rates there during the previous Omicron BA.1 wave (December 2021 - January 2022)⁷. Therefore, the impact of seeding location, immunity from previous waves or vaccination, and NPI changes all contribute to the large and continued spatial heterogeneity in the spread of VOCs.”

Tommy Lam

Referee #3 (Remarks to the Author):

McCrone and colleagues present an large, in-depth investigation into the establishment of the SARS-CoV-2 Delta variant in England in mid-2021. They take advantage of the expansive COG-UK database to capture introductions of Delta into England and track the fate of these introductions. These results have important bearing on non-pharmaceutical interventions as well as how different viral variants fare against preexisting immunity. This latter aspect, however, is the weakest part of the current manuscript.

Authors' response: We thank the reviewer for their positive feedback and comments. Based on the feedback from this and other reviewers, we have removed the results from the US that this reviewer judged to be the weakest part.

The authors do an excellent job exploring and explaining why the actual number of Delta introductions into the UK is likely greater than what they have estimated here. I wonder if there is an additional bias at play that has not been explained: repeated introduction of virus identical (or descendent) genomes. In the absence of synapomorphies distinguishing SARS-CoV-2 genomes from UK and India, a Bayesian phylogeographic analysis would tend group virus from the same country together (without additional information to the contrary). I imagine the migration patterns from the rest of the analysis would inform these probabilities, but my hunch is that, all else being equal, this inference would underestimate the number of introductions when the genomes are uninformative. If this supposition is correct, it would bolster the author's claims that their migration count is indeed biased downwards.

Authors' response: Thank you. Yes, in the absence of genetic information, the location trait would be the deciding factor for grouping sequences. We have to some extent mitigated this bias by performing the phylogenetic and phylogeographic reconstructions as separate steps, i.e. we first infer a posterior set of trees using only genetic information and sampling times (combined with the tree prior means identical genomes are separated by non-zero branches), and then perform a phylogeographic analysis using only the phylogenies from the previous step. However, the phylogeographic analysis would still prefer those phylogenies where sequences from each location are clustered together (the more parsimonious reconstructions with fewer migration events). Thus, we agree that this is probably also an additional contributing factor to underestimating the true number of introductions into the UK. We therefore performed analyses between independent measures of importations (EII) and genomic data which are highly correlated. We have added more detail in the results and discussion: “High variation in sampling intensity among countries (specifically high sampling intensity in England compared to many other countries) means the true number of importations into England is likely much larger than that inferred from phylogeographic analysis alone (Fig. 1b & c, see related discussion in the context of the UK's first wave²).”

The analysis of pre-existing immunity (whose proxy is estimated using the fraction of people with two vaccine doses) is weak. Further, the comparison to the US is not clearly motivated in the text. In much of the US, one could expect an inverse relationship between vaccine uptake and prior infection, due to

regional behavioral difference. Also, public health messaging in the US during the Delta wave did not acknowledge potential for vaccine-breakthrough, so vaccination could have increased risk behavior. Given all the caveats the authors include in this section, they seem to acknowledge these and other weakness in this section. Consider removing.

Authors' response: We agree with the reviewer and have removed data/analyses from the US states. Differences in spatial resolution of the data in addition to those listed by the reviewer mean that we cannot directly compare the US estimates to those from England.

Even if hotel quarantine measures were effective at reducing introductions, it seems the variant was already established in England, rendering moot the impact of restricting additional introduction. Perhaps I misunderstand, but I would appreciate a more explicit interpretation by the authors on the impact of hotel quarantine on the trajectory of the Delta wave in England.

Authors' response: We agree with the reviewer and this is the conclusion we come to in the abstract and discussion: “Subsequently, increased levels of local population mixing, not the number of importations, was associated with faster relative growth of Delta.” & “By pairing the phylogenetic results with contact tracing data we conclude that hotel quarantine measures were effective in reducing onward transmission of imported Delta cases in England. However, after May 21, we found that levels of local social mixing in England, not the number of importations, was associated with faster relative growth of Delta. At that point the independently introduced transmission lineages grew at a similar pace; details of their geographic distribution and expansion will support future work defining the optimal spatial interventions to reduce transmission of VOCs in England.”

This point is also reflected in the title of our paper, which states that the emergence dynamics are “context-specific”, i.e. depend on the properties of the population into which importations occur.

We have added another sentence to the last paragraph reiterating the implications this has for future variants: "Our work highlights the relative importance of local (within country) behavioural and mobility changes in determining the speed at which Delta spread in England; such changes will likely be more important than international travel restrictions during the emergence of future variants."

Minor comments:

Page 3, line 97. ‘Random’ is ambiguous. Do the authors mean to suggest these samples were chosen randomly from all COVID+ samples collected during this time period? Also, since low CT samples are less likely to produce high quality genome sequences, random needs to be qualified.

Authors' response: We apologise for not being more specific. Genome sequencing was performed on a random subset of COVID-19 positive samples with a CT value < 30. During the study period this represented about 40% of all COVID-19 positive samples. We have rewritten the sentence in the introduction which now reads: “Here we examine virus genomes generated from a random sample of all COVID-19 positive tests with CT value below 30 collected during community-based testing in England, between March 12 and June 15, 2021.” We have added additional detail in the Methods section.

Page 3, line 99. ‘Evenly’ should be replaced by ‘uniformly’, since it presumably refers to a uniform distribution.

Authors’ response: We agree with the reviewer and have made the change in the main text.

Page 4, line 143. *Given the sparse sampling in other countries [not India], it seems likely that introductions from low-surveillance countries abroad could be misinterpreted as coming from India. How effective is the travel-aware model (which does an excellent job incorporating travel data) at accounting for this discrepancy in sampling? It’s unclear if the ‘global’ category (page 21) is capturing both well- and poorly-sampled countries.*

Authors’ response: The introductions from low-surveillance countries abroad are explicitly categorized as ‘Global’ sequences, i.e. as not having come from India. As such, sequences are characterized as coming from three key regions of interest (for the purpose of this study): India, England and the rest of the world. Incorporating individual travel histories helps to generate more realistic dispersal patterns, especially when used in conjunction with so-called ‘ghost-lineages’ that account for unsampled taxa from locations that are undersampled (Lemey et al., 2020). It was not possible to include ‘ghost-lineages’ in the current framework (they cannot be used in combination with empirical tree distributions in BEAST) and expect we may mis-identify some early, global introductions as coming from India, as suggested by the reviewer.

However, these inaccuracies are not expected to invalidate our conclusion that the majority of early delta imports into the UK originate from India. This phylogenetic conclusion is further supported by an independent EII analysis, which shows that India dominates global Delta exports for the majority of the study period (Extended Data Figure 2). We also performed analyses to compare the importations labeled as Global and from India from genomic data with importation rates (EII) independently estimated from incidence and mobility data. The two sets of estimates strongly correlate (Extended Data Fig. 4; previous Figure 2c). The correlation between these two independent data sources suggests our conclusions are robust against possible biases present in the phylogenetic analysis.

We added the motivation for doing this analysis in the main manuscript: “To investigate the importation of Delta into England specifically, and to cross-validate the results above using independent data, we estimate the Estimated Importation Intensity (EII) of Delta to England through time^{2,12}. The EII is a metric of Delta importation that represents trends in the number of Delta cases arriving in the country, irrespective of whether or not those cases result in local transmission.”

Page 7, *Regarding the dissemination of Delta in England, it is unclear if urban areas in England were disproportionately sources of dissemination or just areas with large numbers of people.*

Authors’ response: We thank the reviewer for their suggestion. We find that urban areas with higher population sizes tend to have higher numbers of exportations which is expected during an epidemic (Figure, Pearson’s $r = 0.54$, p -value < 0.005). However, locations of early importations and spread had higher than expected numbers of exportations (e.g., Greater Manchester): “Further, we observe that

Bolton, Blackburn, Salford, Bury and Greater Manchester had on average higher than expected numbers of exportations for their population sizes (Extended Data Fig. 9).”

Figure: Number of estimated exportations from phylogeographic analysis and population size at the UTLA level (Pearson’s $r = 0.54$, p -value < 0.005).

Page 11, it would be commenting on how similar these dynamics are to the establishment of SARS-CoV-2 at the start of the pandemic (i.e., >1000 introductions). And perhaps worth commenting on how these dynamics are different.

Authors’ response: We thank the reviewer for their comment and have added additional text to the discussion putting results of the dispersal dynamics of Delta in context of the emergence of SARS-CoV-2 during the first wave, the Alpha wave, and most recent Omicron waves: “In comparison to the Alpha variant, which arose and spread from a single location in South East England¹⁰, the expansion of the Delta variant was predominantly due to exports from the North West (Figure 3). Both the analysis of the Alpha variant and of the first wave of SARS-CoV-2 in spring 2020 suggested that Greater London played an substantial role in spreading SARS-CoV-2 across England, as expected, given it is England’s largest city by far , and highly connected by road, rail and air to other locations. However, Greater London was less important in the spread of Delta, even after Delta had become established there. This indicates the importance of founder effects; where a VOC first becomes established within a country may have a strong impact on subnational spatial dissemination, information that is useful for planning localised interventions.

Further, while there are intrinsic differences in transmissibility between VOCs, the role of NPIs and levels of immunity from prior infection or vaccination also impact their dynamics. After the start of the first UK national lockdown during the first wave of infections in 2020, lineage movements were severely curtailed and most lineages went extinct (Du Plessis et al. 2021); in comparison, viral movement of Delta lineages

increased after the relaxation of NPIs, accompanied with a subsequent rise in positive cases (Figure 3). For the latest VOC, Omicron, NPIs have remained relatively stable throughout England, and the increase in cases of Omicron BA.2 in more rural areas in the South West in February and March 2022 has been speculated to be due to lower infection rates there during the previous Omicron BA.1 wave (December 2021 - January 2022)⁷. Therefore, the impact of seeding location, immunity from previous waves or vaccination, and NPI changes all contribute to the large and continued spatial heterogeneity in the spread of VOCs.”

References

1. Abbott, S. & Funk, S. Estimating epidemiological quantities from repeated cross-sectional prevalence measurements. *bioRxiv* (2022) doi:10.1101/2022.03.29.22273101.
2. du Plessis, L. *et al.* Establishment and lineage dynamics of the SARS-CoV-2 epidemic in the UK. *Science* **371**, 708–712 (2021).
3. Lo, A., Chernoff, H., Zheng, T. & Lo, S.-H. Why significant variables aren’t automatically good predictors. *Proc. Natl. Acad. Sci. U. S. A.* **112**, 13892–13897 (2015).
4. Tian, H. *et al.* An investigation of transmission control measures during the first 50 days of the COVID-19 epidemic in China. *Science* **368**, 638–642 (2020).
5. GISAID - Initiative. <https://www.gisaid.org/>.
6. Gelman, A. & Stern, H. The Difference Between ‘Significant’ and ‘Not Significant’ is not Itself Statistically Significant. *Am. Stat.* **60**, 328–331 (2006).
7. Elliott, P. *et al.* Twin peaks: the Omicron SARS-CoV-2 BA.1 and BA.2 epidemics in England. (2022).
8. [No title].
https://assets.publishing.service.gov.uk/government/uploads/system/uploads/attachment_data/file/1001354/Variants_of_Concern_VOC_Technical_Briefing_17.pdf.
9. Booking and staying in a quarantine hotel if you’ve been in a red list country.
<https://www.gov.uk/guidance/booking-and-staying-in-a-quarantine-hotel-when-you-arrive-in-england>.
10. Kalkauskas, A. *et al.* Sampling bias and model choice in continuous phylogeography: Getting lost on

a random walk. *PLoS Comput. Biol.* **17**, e1008561 (2021).

11. Du Plessis, L., McCrone, J. T., Zarebski, A. E., Hill, V. & Ruis, C. Establishment and lineage dynamics of the SARS-CoV-2 epidemic in the UK. *Science* (2021).
12. Kraemer, M. U. G. *et al.* Spatiotemporal invasion dynamics of SARS-CoV-2 lineage B.1.1.7 emergence. *Science* **373**, 889–895 (2021).

Reviewer Reports on the First Revision:

Referees' comments:

Referee #1 (Remarks to the Author):

I am happy with the response of the authors to my previous comments and suggestions.

Chris Illingworth

Referee #2 (Remarks to the Author):

The authors have fully addressed my comments in previous review. I have only minor comments on some unclarity that I did not spot previously:

Figure 3c. May add title for the x-axis in addition to just 'km'.

Figure 3d. In the legend text, it states "see transmission lineages IV-VII in Extended Data Fig. 7". I believe it is more precise by saying "IV, VI and VII", instead of "IV-VII".

Referee #3 (Remarks to the Author):

I am satisfied with the revised version of the manuscript. My concerns have been thoroughly addressed.

Author Rebuttals to First Revision:

Response: “Figure 3c. May add title for the x-axis in addition to just ‘km’.”

We have changed the title to “Virus lineage movement distance (km)”

“Figure 3d. In the legend text, it states “see transmission lineages IV-VII in Extended Data Fig. 7”. I believe it is more precise by saying “IV, VI and VII”, instead of “IV-VII”.” We have made this change to the legend.